# ZIP1$^+$ fibroblasts protect lung cancer against chemotherapy via connexin-43 mediated intercellular Zn$^{2+}$ transfer

Chen Ni [1] ✉, Xiaohan Lou[1], Xiaohan Yao[1], Linlin Wang[1], Jiajia Wan[1], Xixi Duan[1], Jialu Liang[1], Kaili Zhang[1], Yuanyuan Yang[1], Li Zhang[1], Chanjun Sun[1], Zhenzhen Li[1], Ming Wang[1], Linyu Zhu[1], Dekang Lv [2] ✉ & Zhihai Qin [1,3] ✉

Tumour–stroma cell interactions impact cancer progression and therapy responses. Intercellular communication between fibroblasts and cancer cells using various soluble mediators has often been reported. In this study, we find that a zinc-transporter (ZIP1) positive tumour-associated fibroblast subset is enriched after chemotherapy and directly interconnects lung cancer cells with gap junctions. Using single-cell RNA sequencing, we identify several fibroblast subpopulations, among which $Zip1^+$ fibroblasts are highly enriched in mouse lung tumours after doxorubicin treatment. ZIP1 expression on fibroblasts enhances gap junction formation in cancer cells by upregulating connexin-43. Acting as a Zn$^{2+}$ reservoir, ZIP1$^+$ fibroblasts absorb and transfer Zn$^{2+}$ to cancer cells, leading to ABCB1-mediated chemoresistance. Clinically, ZIP1$^{high}$ stromal fibroblasts are also associated with chemoresistance in human lung cancers. Taken together, our results reveal a mechanism by which fibroblasts interact directly with tumour cells via gap junctions and contribute to chemoresistance in lung cancer.

Lung cancer is the leading cause of cancer death worldwide[1]. Chemotherapy, radiotherapy, targeted biological therapy, and in recent years immunotherapy are central treatment options for lung cancer patients[2]. Chemotherapy failure is a major challenge in the clinical treatment of lung cancer, and better therapeutic strategies are urgently needed[2]. Chemotherapy is often prescribed to patients with early-stage cancer as an adjunct therapy for surgery, or to patients with advanced cancer to suppress tumour formation[2]. After an initial response, relapse and metastasis commonly occur due to drug resistance, and patients usually succumb to the disease. Unfavourable tumour–stoma crosstalk may contribute to the chemoresistance of lung cancer by modulating the extracellular matrix, generating an immunosuppressive niche, and promoting cancer stem cells[3,4]. When treated with drugs, stromal fibroblasts can adapt to chemotherapy-induced damage and communicate with cancer cells to affect drug responses[5,6]. Understanding fibroblast-tumour communication will undoubtedly be helpful for the improvement of lung cancer therapy.

Cancer-associated fibroblasts (CAFs) are heterogeneous stromal cells that are implicated in lung cancer chemoresistance[4,7]. Long-range interactions by secreting factors or exosomes between CAFs and tumours have been extensively studied which impact tumour progression and therapy responses[8]; but short-range interactions, including cell–cell contacts between CAFs and tumours, have rarely been described[9]. We and others have reported that fibroblasts can directly link with tumour cells to lead to their metastasis[10,11]. A previous study reported that CAFs may form gap junctions with lung cancer cells to exchange metabolites[12]. Gap junctions are intercellular communication channels that connect neighbouring cells, allowing the passage of small molecules (<1000 Da)[13]. The membrane protein connexin forms a hexameric structure as a connexon that docks with a

[1]Medical Research Center, the First Affiliated Hospital of Zhengzhou University, 450052 Zhengzhou, China. [2]Institute of Cancer Stem Cell, Dalian Medical University, 116044 Dalian, China. [3]Institute of Biophysics, Chinese Academy of Sciences, 100101 Beijing, China. ✉e-mail: nichen904@163.com; dekanglv@126.com; zhihai@ibp.ac.cn

corresponding connexon on a neighbouring cell to form a gap junction channel[13]. It has been reported that brain metastases of lung cancer cells link astrocytes by gap junctions and activate astrocytes by cGAMP transfer, thereby supporting tumour growth and chemoresistance[14]. However, the regulatory mechanisms of fibroblast–cancer cell intercellular communication by gap junctions during chemotherapy is still unknown.

Here, we show ZIP1+ CAF subset is enriched in lung cancer models after chemotherapy and actively transfers $Zn^{2+}$ to cancer cells, promoting ABCB1-mediated chemoresistance. In this study, we use single-cell RNA sequencing (scRNA-seq) to examine the adaptations of fibroblasts after chemotherapy in a mouse lung cancer model. We identify a zinc-transporter positive (Zip1+) CAF subset, which is enriched in mouse lung cancer treated with chemotherapy. Zrt- and Irt-like protein 1 (ZIP1) (also known as SLC39A1) is a zinc transporter that imports $Zn^{2+}$ from the extracellular space[15]. Unexpectedly, we find that ZIP1+ fibroblasts interconnect cancer cells via gap junctions by upregulating connexin-43 (CX43). Furthermore, ZIP1+ fibroblasts act as $Zn^{2+}$ reservoirs to absorb and transfer $Zn^{2+}$ to lung cancer cells, inducing ABCB1-mediated drug extrusion. Finally, the contribution of ZIP1+ fibroblasts to chemoresistance in lung cancer cells is studied. Our findings provide insights into fibroblast–tumour short-range interactions and have important implications for lung cancer therapy.

## Results

### ZIP1+ fibroblasts are enriched in mouse lung cancer after chemotherapy

We aimed to study the early-stage adaptation of CAFs to chemotherapy-induced damage. Considering that lung cancer patients receiving chemotherapy are usually at advanced stages with distant metastases, the ectopic tumour model might facilitate the exploration of potential origin-nonspecific changes in CAFs. Lewis lung carcinoma (LLC) cells were subcutaneously transplanted into C57BL/6 mice with ubiquitous enhanced green fluorescent protein (EGFP) expression. The established tumours (-100 mm³) were treated with doxorubicin (DOX) three times every 2 days. Two days after treatment cessation, EGFP+ CD45− tumour stromal cells, excluding immune cells (CD45+), were isolated and sorted for scRNA-seq (Fig. 1a, b, Supplementary Fig. 1a, b). Eight stromal cell clusters were identified (Fig. 1c, d, Supplementary Table 1; Supplementary Fig. 1c–e). Cluster 5 cells are typical endothelial cells that express Pecam1, Cldn5 and Plvap. Cluster 4 cells expressed genes encoding monocyte markers (CD11b encoded by Itgam, Lyz2, CD14) and fibroblast markers (Dcn, Fn1, Sparc and Opn encoded by Spp1), implying that they are likely bone marrow-derived fibrocytes[16]. A recent study also identified fibroblasts expressed both fibroblast and macrophage markers and proposed macrophage–myofibroblast transition mechanism for the generation of these cells[17]. Clusters 0–3 (>95% of the cells) and 6–7 were non-epithelial, non-endothelial, and non-hematopoietic fibroblasts, expressing fibroblast markers (Dcn, Fn1, Sparc and Opn). Interestingly, we found that the Cluster 0 fibroblast subset highly expressed Zip1 (encoded by slc39a1) (Fig. 1c, d, Supplementary Table 1), a $Zn^{2+}$ transporter that imports $Zn^{2+}$ from the extracellular space to the cytosol[15]. ZIP1 belongs to the solute carrier family SLC39, which is responsible for maintaining cytosolic zinc concentrations[15]. The human SLC39 family includes members ZIP1–14, among which ZIP4, ZIP6, ZIP7, ZIP10, and ZIP14 have been implicated in cancer progression, cachexia, and immunodeficiency by tuning cellular $Zn^{2+}$ and zinc-associated signalling[18–21]. Compared to tissues from untreated control mice, Cluster 0 Zip1+ CAF cells were increased in tumours from DOX-treated mice, whereas other CAF subsets were decreased (Fig. 1e, f).

The biological function of each CAF subset was explored (Fig. 1g, Supplementary Data 1). According to gene-set variation analysis (GSVA) of 50 hallmark pathways, small Cluster 6 and 7 enriched "myogenesis" signatures and produced extracellular matrix (ECM)

(Fig. 1g, Supplementary Fig. 1f), which are representative myofibroblasts (myCAF)[7,22]. The "myogenesis" signature was also active in Cluster 2 CAFs (Fig. 1g). Notably, Cluster 2 Col3a1+ CAFs expressed Nnmt, which encodes nicotinamide N-methyltransferase (Supplementary Fig. 1g), which has been associated with a "hypomethylation" phenotype of CAFs (metCAFs)[23]. Cluster 3 Hmgb2+ CAFs expressed proliferation marker genes Cdk1, Mki67, Ccna2, Ccnb1 and Ccnb2 (Supplementary Table 1), and enriched the pathways "mitotic spindle", "G2M checkpoint" and "Myc targets" (Fig. 1g). No specific positive markers were identified for Cluster 1 CAFs (Supplementary Table 1). Cluster 0 Zip1+ CAFs expressed the marker genes Notch2 and Zip1 (Supplementary Table 1), which are associated with embryonic development (Supplementary Data 1), implying a developmental CAF phenotype[24]. Notch2 has been reported to be critical for the maintenance of cell stemness in hematopoietic cells and neural stem cells[25,26]. Cluster 0 Zip1+ CAFs moderately expressed Dpt (Supplementary Fig. 1h), which labels universal fibroblasts[27]. We further applied the cNMF algorithm to the cells of Cluster 0-3 to identify gene expression programmes (GEP) in these cells[28]. We identified the top 100 genes for each GEP (Supplementary Table 2) and performed a GO (Gene Ontology) analysis to explore the function of each GEP (Supplementary Data 2). Relating to the clusters and programmes, we found that of the four clusters, Cluster 0 was strongly enriched for developmental GEP1, Cluster 1 was strongly enriched for metabolic GEP2, Cluster 2 was strongly enriched for matrix GEP3, and Cluster 3 was strongly enriched for proliferating GEP4, further supporting that they are distinct clusters (Supplementary Fig. 1i). Although there were no specific markers, Cluster 1 highly expressed metabolism-related genes, such as Gchfr, Mif, and Cox8a, that were shared with other clusters (Supplementary Fig. 1j). To compare pathway activity among the seven clusters, we used PROGENy to estimate the activity of the 14 pathways[29]. The PI3K signalling pathway was active in Cluster 0 Zip1+ CAFs, while the TGFβ signalling pathway was active in Cluster 6 and 7 myofibroblasts, the JAK-STAT signalling pathway was active in Cluster 4 fibrocytes, and the VEGF signalling pathway was active in Cluster 2 (Supplementary Fig. 1k). It has been reported that TGFβ stimulates myofibroblast phenotype and IL-1 (activating JAK-STAT) promotes inflammatory fibroblasts[7,30]. Therefore, Cluster 0 Zip1+ CAFs might be driven by a distinct PI3K signalling pathway compared to other clusters.

RNA velocity analysis was performed to evaluate the fate of fibroblasts. Two strong differentiation routes were observed in the DOX group. One route is from Cluster 0 to Cluster 2, and the other is within Cluster 0 (Fig. 1h), which implies that Cluster 0 cells tend to partially differentiate into metCAFs and partially retain their phenotype. We evaluated gene upregulation in cluster transition. For example, Plod2, Nnmt and Col3a1 were upregulated in transition from Cluster 0 to Cluster 2, while Spry2, Mt2 and Mt1 were upregulated in transition within Cluster 0 (Supplementary Fig. 1l–o, Supplementary Table 3). Immunostaining confirmed that DOX treatment elevated the number of ZIP1+ fibroblasts compared to control LLC tumours (Supplementary Fig. 1p, q). In another transplanted tumour model of pancreatic cancer Pan02, we also observed the presence of ZIP1+ fibroblasts (Supplementary Fig. 1r), implying that ZIP1+ fibroblasts may be a common fibroblast subpopulation. Moreover, in human lung adenocarcinoma-associated fibroblasts (hCAFs) isolated from lung adenocarcinoma with neoadjuvant chemotherapy, ZIP1+ fibroblasts were present (Supplementary Fig. 1s).

### ZIP1+ fibroblasts interconnect lung cancer cells by upregulating CX43

These RNA velocity results imply that Cluster 0 may contain two subgroups of fibroblasts. We tuned the resolution of single-cell RNA analysis and separated Cluster 0 ZIP1+ fibroblasts into two subgroups (Cluster CA0 and CA1) (Fig. 2a). Compared with the PBS group, CA0 was the most increased cell subset, followed by CA1, in the DOX group

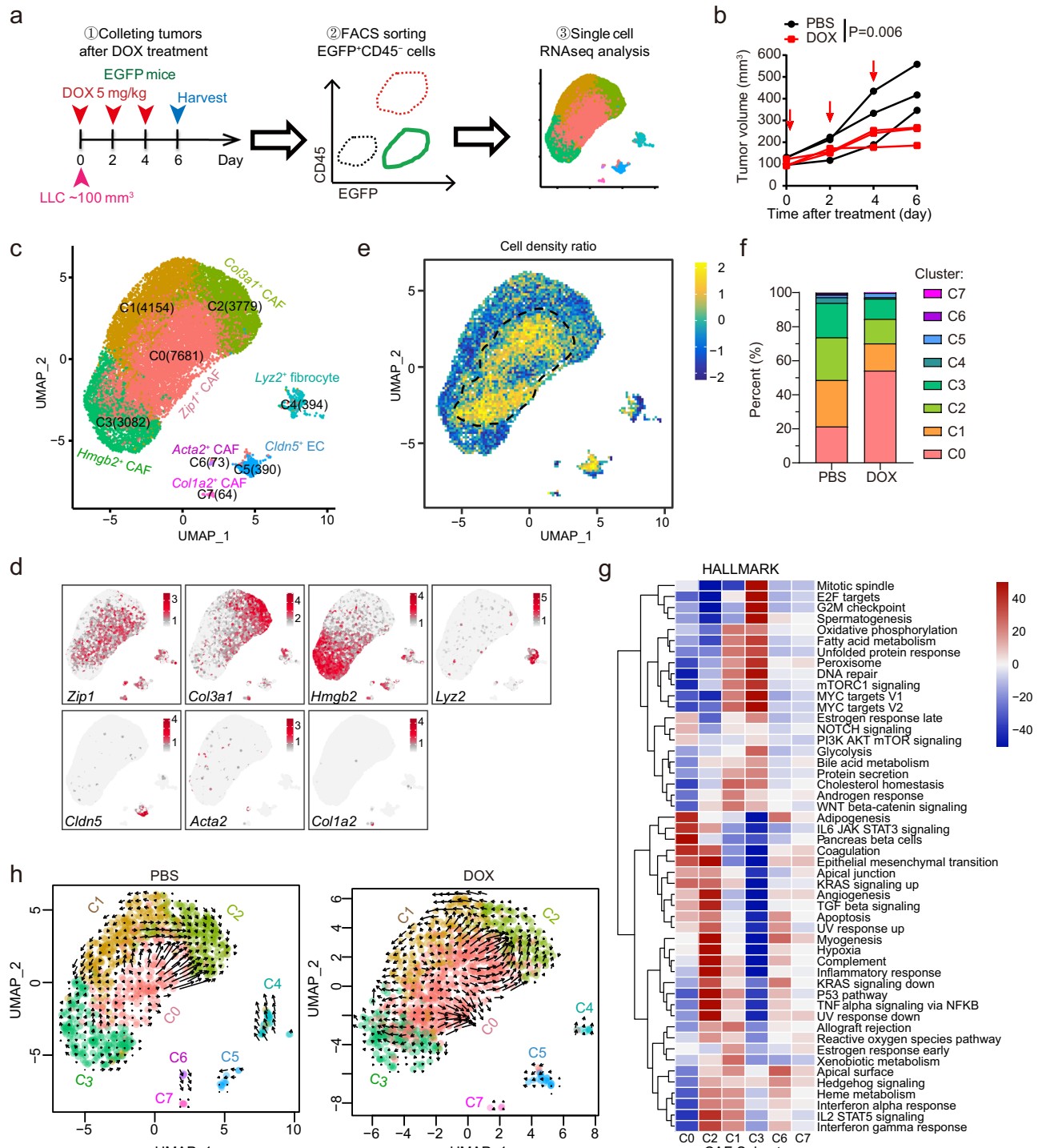

**Fig. 1 | ScRNAseq reveals enrichment of *Zip1*+ fibroblasts in LLC tumours following DOX treatment. a** Schematic of experimental design for single-cell RNA sequencing (scRNAseq) of stromal fibroblasts after chemotherapy in LLC tumour transplanted into C57BL/6-EGFP mice. **b** Tumour growth curve of LLC tumours treated with or without doxorubicin (DOX) as described in **a**. *n* = 3 for each group. Two-way ANOVA test for tumour growth curves. **c** UMAP plot displaying 19,617 cells from six mice (three treated with DOX and three PBS-treated controls), indicating the numbers of cells in each of the eight subsets. **d** UMAP plots showing the expression of subtype-specific marker genes. **e** UMAP plot showing the relative cell density ratio for DOX vs. PBS treatment. **f** Faceted bar graph showing the percentages of cells in each subset with PBS and DOX treatment. **g** Heatmap of gene-set variation analysis of hallmark signalling pathway terms in the six fibroblast subsets. **h** UMAP plots of RNA velocity indicating cell-fate transitions. Source data are provided as a Source Data file (**b**, **f**).

(Fig. 2a, b). CA0 and CA1 highly expressed *Zip1* while weakly expressing the other zinc transporters (Supplementary Fig. 2a). In addition, CA0 cells expressed higher *S100a4* when compared to CA1 cells (Supplementary Fig. 2b). To discriminate the specific functions of CA0 and CA1, each subtype was compared with all other subtypes within CA0–7

and pathway enrichment was performed. The results showed that CA0 enriched "gap junction" while CA1 enriched "cell cycle" and "apoptosis" (Fig. 2c).

Next, we examined whether the expression of ZIP1 in fibroblasts affects gap junction protein expression. Connexin-43 (CX43)

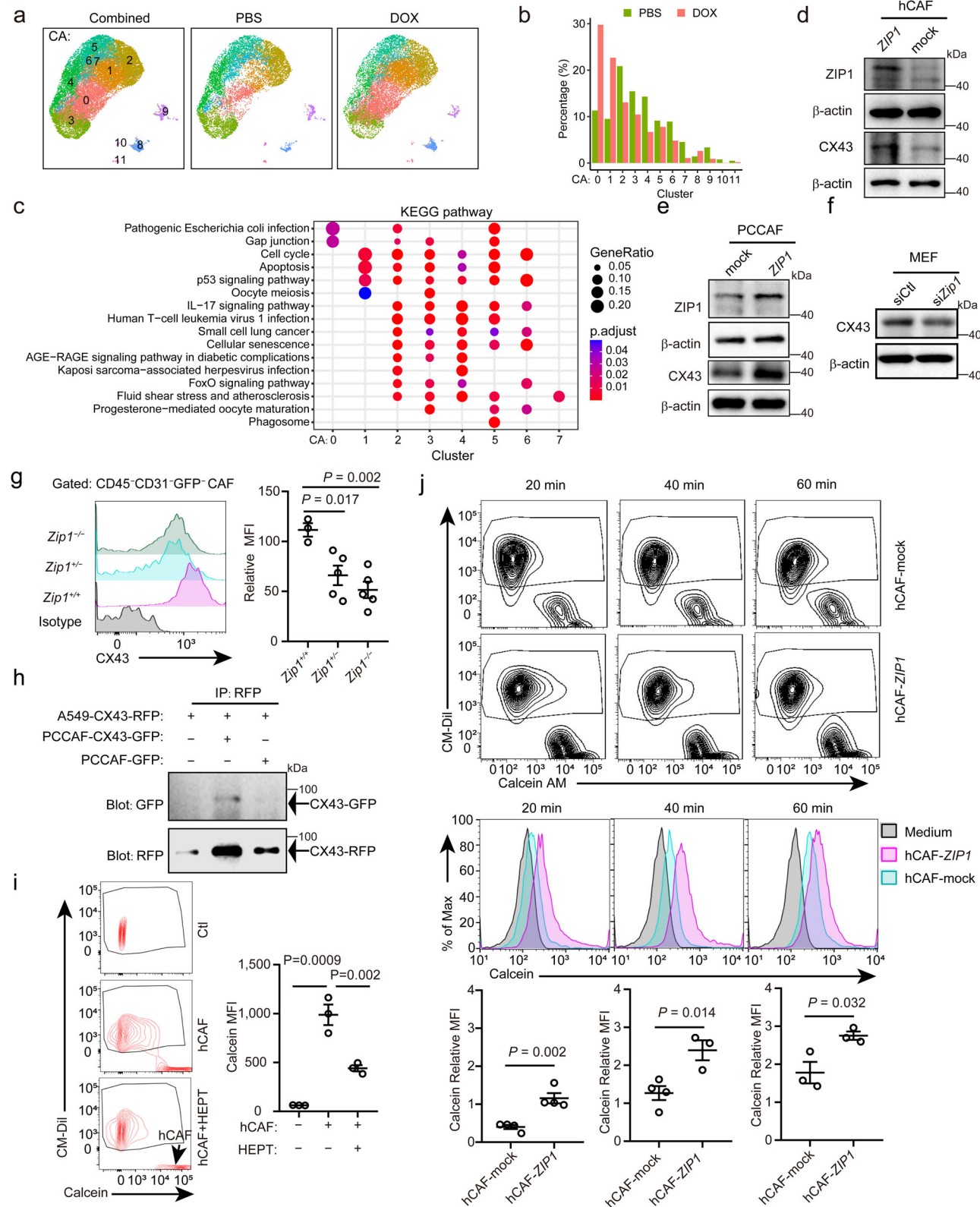

expression has been observed in fibroblasts and engages in myocyte–fibroblast coupling[31,32]. The results showed that overexpression of ZIP1 in hCAF increased the expression of CX43 (Fig. 2d, Supplementary Fig. 2c). Moreover, overexpression of ZIP1 in prostate cancer-associated fibroblasts (PCCAFs) and mouse embryonic fibroblasts (MEFs) increased the expression of CX43 (Fig. 2e, Supplementary Fig. 2d, e), implying that the regulation of CX43 by ZIP1 is

independent of fibroblast origin. Conversely, knockdown of *Zip1* in MEFs decreased the expression of CX43 (Fig. 2f, Supplementary Fig. 2f). To further confirm the ZIP1-dependent CX43 expression in lung CAF, we transplanted LLC-GFP-luc tumour cells into *Zip1*^+/+, *Zip1*^+/– and *Zip1*^–/– mice. On day 22, the expression of CX43 on CD45⁻CD31⁻GFP⁻ CAF was analysed by flow cytometry. CD45⁻CD31⁻GFP⁻ CAF from *Zip1*^+/+ mice expressed higher levels of

**Fig. 2 | ZIP1⁺ fibroblasts interconnect cancer cells with gap junctions by upregulating CX43. a** UMAP plot showing 12 cell subsets in PBS group, DOX group and their combined. **b** Bar plots showing the percentages of cells in each subset with PBS and DOX treatment. **c** Gene-set variation analysis of KEGG signalling pathway terms within the 8 fibroblast subsets (CA0–7). **d, e** ZIP1 and CX43 expression in **d** human lung adenocarcinoma-associated fibroblast (hCAF) and **e** prostate-cancer cancer-associated fibroblast (PCCAF) transfected with control (mock) or ZIP1-overexpression vector (*ZIP1*). Representative results from four (**d**), three (**e**, ZIP1) or five (**e**, CX43) independent experiments are shown. **f** CX43 expression in mouse embryonic fibroblasts (MEFs) transfected with control (siCtl) or *Zip1*-silencing (si*Zip1*) siRNA. A representative result from five independent experiments is shown. β-actin was used as control. The samples derived from the same experiment and blots were processed in parallel (**e-f**). **g** CX43 expression on the cell surface of *Zip1*⁺/⁺, *Zip1*⁺/⁻ and *Zip1*⁻/⁻ CAFs. Single-cell suspensions of LLC-GFP-luc tumours from indicated mice were analysed by FACS on day 22. CD45⁻CD31⁻GFP⁻ CAFs were gated. Relative mean fluorescence intensity (MFI) was

calculated as MFI[sample]/MFI[negative control]−1. Data are presented as mean ± SEM, *n* = 3 for *Zip1*⁺/⁺, *n* = 5 for *Zip1*⁺/⁻ and *Zip1*⁻/⁻. Two-tailed *t* test. **h** Direct CX43-CX43 interaction between A549 and PCCAF. A549-CX43-RFP and PCCAF-CX43-GFP were constructed and cocultured. Co-immunoprecipitation (IP) of CX43-RFP and CX43-GFP was examined by western blotting. A representative result from two independent experiments is shown. (**i**) Calcein transfer from hCAFs to A549 tumour cells. A549 cells (pre-labelled with CM-Dil) and hCAFs (loaded with 0.5 μM calcein-AM) were co-cultured for 2 h in DMEM + 10% FBS. Heptanol (HEPT, 2 mM) was used to block gap junctions. Calcein fluorescence in tumour cells was determined by FACS. Arrow, the position of hCAFs. Mean ± SEM, *n* = 3 for each group. Two-tailed *t* test. **j** Calcein transfer from hCAF-mock/hCAF-*ZIP1* to A549 tumour cells at the indicated time. A549 alone without fibroblasts were used as a control. Calcein relative MFI in A549 cells of different groups was compared. Mean ± SEM, *n* = 4 for hCAF-mock 20, 40 min and hCAF-*ZIP1* 20 min, *n* = 3 for hCAF-mock 60 min and hCAF-*ZIP1* 40, 60 min. Two-tailed *t*-test. Source data are provided as a Source Data file (**b, d–j**).

membranous CX43 than those from *Zip1*⁺/⁻ and *Zip1*⁻/⁻ mice (Fig. 2g, Supplementary Fig. 2g). Consistent with this observation, *Zip1*⁺/⁻ MEF expressed intermediate levels of *Zip1* mRNA compared to *Zip1*⁺/⁺ MEF and *Zip1*⁻/⁻ MEF (Supplementary Fig. 2h).

Next, we examined whether CX43 on CAFs could physically interact with CX43 in tumour cells. We ectopically expressed CX43-GFP in PCCAFs and CX43-RFP in A549 cells, which were physically in contact with each other (Supplementary Fig. 2i). Furthermore, CX43-GFP co-immunoprecipitated with CX43-RFP in lysates of PCCAF-CX43-GFP and A549-CX43-RFP co-culture (Fig. 2h), indicating a CX43-CX43 physical interaction between CAFs and tumour cells. The results further showed that functional gap junctions were formed between A549 tumour cells and hCAFs, as indicated by the transfer of the fluorescent dye calcein from fibroblasts to tumour cells, which could be impaired by heptanol, the gap-junction uncoupler (Fig. 2i). In co-cultures, ZIP1-overexpressing hCAFs and PCCAFs were more efficient in transferring calcein to A549 tumour cells than to control fibroblasts (Fig. 2j, Supplementary Fig. 2j). Upon CX43 knockdown, the transfer of calcein by hCAFs was impaired (Supplementary Fig. 2k). These results demonstrated that ZIP1⁺ fibroblasts form efficient gap junctions with tumour cells through the upregulation of CX43.

## ZIP1⁺ fibroblasts act as a Zn²⁺ reservoir and transfer Zn²⁺ to cancer cells

Interconnections between fibroblasts and cancer cells via gap junctions enable electrical and metabolic exchanges between cells. Considering Zn²⁺ as a substrate of ZIP1, we explored the possible shuttle transport of Zn²⁺ between the two cells. The results showed that after DOX treatment, labile Zn²⁺ levels in the tumour interstitial fluid increased (Fig. 3a). In addition, large necrosis was observed in tumours treated with DOX (Fig. 3b). Thus, we suspected that cell death during chemotherapy may have released intracellular Zn²⁺ into the interstitial space. Notably, the fluorescence of DOX in the culture medium may interfere with the detection of Zn²⁺ using the probe FluoZin3. To validate and generalise the above speculation, we used extreme temperatures (95 °C, −80 °C) and paclitaxel to induce cell death in LLC cells. When tumour cells were treated as described above, Zn²⁺ was released from dying tumour cells, regardless of the type of treatment (Fig. 3c, d).

The results further showed that CAFs were able to absorb Zn²⁺ released by dying cancer cells treated with paclitaxel (Fig. 3e), and the absorption was proportional to the extracellular doses of Zn²⁺ (Fig. 3f). The deficiency of *Zip1* in MEFs or mouse LLC tumour CAFs (mCAFs) reduced the uptake of Zn²⁺ from the extracellular space (Fig. 3g, Supplementary Fig. 3a). Overexpression of *Zip1* in hCAFs and PCCAFs increased the uptake of Zn²⁺ from the extracellular space (Fig. 3h, Supplementary Fig. 3b).

Next, we examined whether fibroblasts could deliver labile Zn²⁺ to cancer cells via the gap junctions. First, labile Zn²⁺ in CAF and tumour cells was determined. The results showed that hCAFs maintained a higher labile Zn²⁺ gradient than A549 tumour cells under both basal (Fig. 3i) and high Zn²⁺ conditions (30 μM) (Supplementary Fig. 3c). Co-cultured with hCAFs increased labile Zn²⁺ in A549 cells, which could be blocked by heptanol or TPEN (a Zn²⁺ chelator) pre-treatment of hCAFs to reduce the Zn²⁺ gradient (Fig. 3j). In Hanks' balanced salt solution (HBSS, with CaCl₂), hCAFs overexpressing ZIP1 effectively transferred Zn²⁺ to A549 cells, and this transfer was blocked by heptanol (Fig. 3k). Notably, control hCAFs could not transfer Zn²⁺ to tumour cells under these conditions (Fig. 3k). *Zip1*⁺/⁺ MEFs, but not *Zip1*⁺/⁻ or *Zip1*⁻/⁻ MEFs, significantly enhanced Zn²⁺ levels in LLC (Fig. 3l). These results demonstrated that ZIP1⁺ fibroblasts can act as Zn²⁺ reservoirs to absorb and transfer Zn²⁺ to tumour cells through gap junctions.

## Zn²⁺ mediates upregulation of CX43 through activating AKT

Because Zn²⁺ can be absorbed by fibroblasts from the extracellular space, we investigated whether Zn²⁺ is involved in the regulation of CX43. Interestingly, the results showed that Zn²⁺ stimulated the upregulation of CX43 and activation of pAKT in a dose-dependent manner in mCAFs and hCAFs (Fig. 4a, Supplementary Fig. 4a, b). This is consistent with the above results showing enrichment of PI3K/AKT signalling in ZIP1⁺ cells (Fig. 1g). The addition of Zn²⁺ led to a reduction in PTEN (an endogenous inhibitor of the PI3K/AKT pathway) in mCAFs (Fig. 4b, Supplementary Fig. 4c). Inhibition of AKT with LY294002 impaired Zn²⁺-induced upregulation of CX43 in mCAFs (Fig. 4c, Supplementary Fig. 4d). Compared to *Zip1*⁻/⁻ MEFs, in *Zip1*⁺/⁺ MEFs AKT was activated and PTEN was downregulated (Fig. 4d, Supplementary Fig. 4e).

These results demonstrated that Zn²⁺ could be transferred to cancer cells by fibroblasts, and the regulation of CX43 in cancer cells by Zn²⁺ was studied. Interestingly, co-culture of LLC-GFP-luc cells with mCAFs resulted in higher expression of CX43 than that in mono-cultured tumour cells, which was attenuated by heptanol or by pre-treatment of fibroblasts with TPEN (Fig. 4e, Supplementary Fig. 4f). When ZnCl₂ was directly added to LLC cell cultures, intracellular labile Zn²⁺ increased in a dose-dependent manner (Supplementary Fig. 4g). Zn²⁺ stimulation resulted in the time-dependent and dose-dependent elevation of CX43 expression in tumour cells (Fig. 4f, g, Supplementary Fig. 4h). Co-culture with *Zip1*⁺/⁺ MEFs, but not with *Zip1*⁻/⁻ MEFs, induced CX43 expression in the LLC-GFP-luc cells (Fig. 4h). In accordance with the above results, CX43 expression was higher on the cell surface of *Zip1*⁺/⁺ MEFs than in *Zip1*⁻/⁻ MEFs (Fig. 4h). These results suggest that Zn²⁺ derived from fibroblasts further increases gap junction formation by upregulating CX43 in cancer cells.

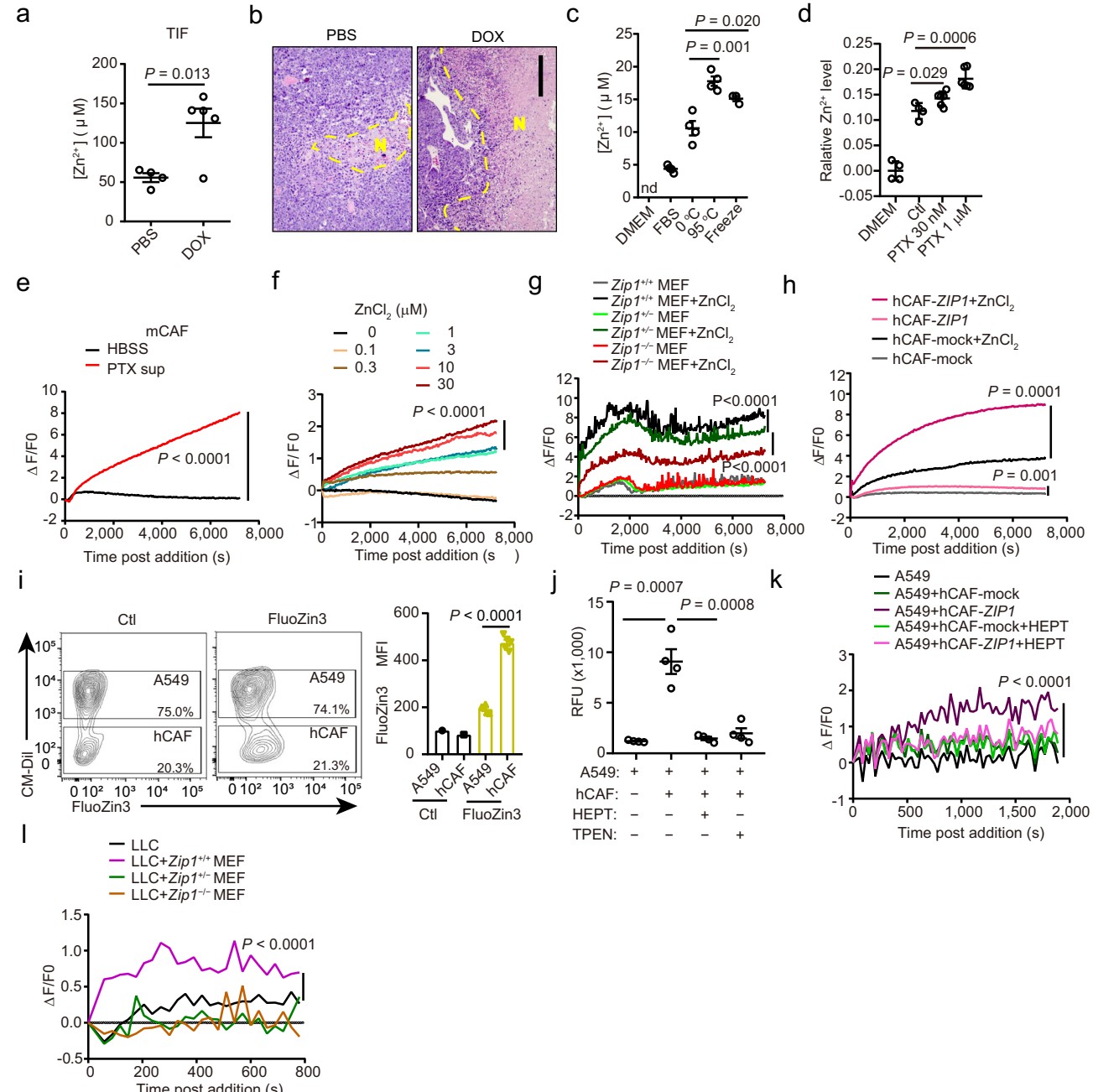

**Fig. 3 | ZIP1+ fibroblasts absorb and transfer Zn2+ to cancer cells through gap junctions. a** Zn2+ in tumour interstitial fluid (TIF) from mice with PBS ($n = 4$) and doxorubicin (DOX, $n = 5$) treatments. Mean ± SEM, Two-tailed $t$-test. **b** Necrosis in tumours with DOX treatment. N necrosis. Scale bar, 200 μm. A representative result from five independent experiments is shown. **c** Zn2+ released from dying tumour cells. LLC cells ($1 \times 10^7$) in 50 μL culture medium were treated at 0 °C,120 min, 95 °C, 15 min, or −80 °C, 60 min (freeze). nd, not detectable. Mean ± SD, $n = 4$. Two-tailed $t$-test. **d** Relative Zn2+ levels of PTX-treated LLC cells. Mean ± SD, $n = 4$ for DMEM and Ctl groups. $n = 6$ for PTX groups. Two-tailed $t$-test. **e** Zn2+ absorption by mCAF from the supernatant of PTX treated LLC cells. $n = 3$. **f** Zn2+ absorption by mCAF from extracellular space containing ZnCl2 at indicated concentrations. $n = 3$. **g** Zn2+ uptake by Zip1+/+, Zip1+/− and Zip1−/− MEFs. ZnCl2, 30 μM.

$n = 3$. **h** Zn2+ uptake by hCAF-mock or hCAF-ZIP1. $n = 3$. **i** Labile Zn2+ in hCAFs and A549 tumour cells (CM-Dil-labelled) co-cultured in DMEM + 10% FBS for 3 h. Fluo-Zin3 was used to detect labile Zn2+ in cells. Ctl, no FluoZin3 staining. Mean ± SEM, $n = 5$. Two-tailed $t$-test. **j** Labile Zn2+ in A549 tumour cells (preloaded with FluoZin3-AM) co-cultured with hCAFs under indicated conditions for 2 h in DMEM. TPEN: 50 μM, pre-treating hCAFs for 5 min. HEPT: heptanol, 2 mM. FluoZin3 fluorescence was read in HBSS. Mean ± SEM, $n = 4$. Two-tailed $t$-test. **k** Labile Zn2+ in A549 tumour cells (preloaded with FluoZin3-AM) co-cultured with hCAF-mock or hCAF-ZIP1 in HBSS. HEPT: 2 mM. $n = 3$. **l** Labile Zn2+ in LLC tumour cells (preloaded with FluoZin3-AM) co-cultured with Zip1+/+, Zip1+/− or Zip1−/− MEFs in HBSS. $n = 3$. Two-way ANOVA test for curve comparison. Source data are provided as a Source Data file (**a**, **c**–**l**).

## Zn2+ induces ABCB1-mediated drug extrusion of cancer cells

The effect of Zn2+ derived from fibroblasts on the tumour response to drugs was studied. We first noted that Zn2+ uptake by tumour cells was suppressed by chemotherapy treatment (Fig. 5a), implying that they might need Zn2+ supplementation from fibroblasts under stress. DOX

treatment-induced drug accumulation (determined by DOX fluorescence in individual cells) and DNA damage (indicated by phosphorylated histone γ-H2AX expression) in tumour cells, which could be attenuated by the addition of fibroblasts to the cultures (Fig. 5b). Interestingly, the addition of heptanol or pretreatment of fibroblasts

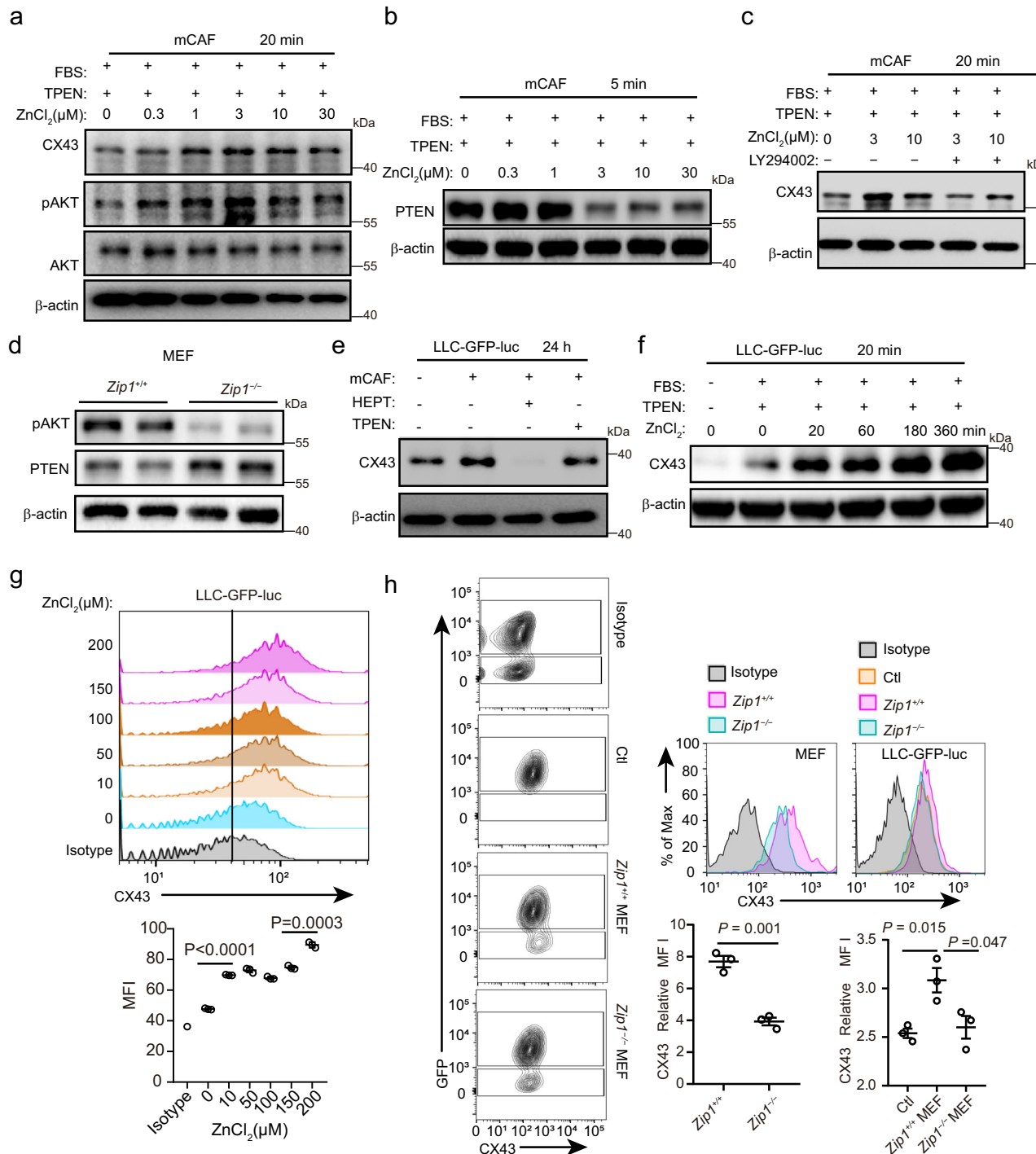

**Fig. 4 | Labile Zn²⁺ upregulates CX43 by modulation of the PTEN/AKT pathway.**
**a** Expression of CX43, pAKT and AKT in mCAFs stimulated with different concentrations of ZnCl₂ for 20 min in DMEM + 10%FBS + 5 μM TPEN. β-actin was used as control. The samples derived from the same experiment and blots were processed in parallel. A representative result from three independent experiments is shown. **b** Expression of PTEN in mCAFs stimulated with different concentrations of ZnCl₂ for 5 min in DMEM + 10%FBS + 5 μM TPEN. A representative result from three independent experiments is shown. **c** Expression of CX43 in mCAFs stimulated with ZnCl₂ ± AKT inhibitor (50 μM LY294002) for 20 min. A representative result from three independent experiments is shown. **d** Expression of pAKT and PTEN in *Zip1⁺/⁺* and *Zip1⁻/⁻* mouse embryonic fibroblasts (MEFs). β-actin was used as a control. The samples derived from the same experiment and blots were processed in parallel. A representative result from three independent experiments is shown. **e** Expression of CX43 in LLC-GFP-luc cells co-cultured with mCAFs. Cells were cultured in DMEM + 10%FBS for 24 h. mCAFs were pre-treated with TPEN (50 μM). HEPT; 2 mM.

β-actin was used as control. The samples derived from the same experiment and blots were processed in parallel. A representative result from four independent experiments is shown. **f** Expression of CX43 in LLC-GFP-luc cells stimulated with ZnCl₂ (30 μM) for different times under indicated conditions. β-actin was used as control. The samples derived from the same experiment and blots were processed in parallel. A representative result from three independent experiments is shown. **g** Expression of CX43 on the cell surface of LLC-GFP-luc tumour cells treated with different concentrations of ZnCl₂ for 4 h in DMEM + 10%FBS + 5 μM TPEN. Data are presented as mean ± SEM, *n* = 3 for each group. Two-tailed *t*-test. **h** Expression of CX43 on the cell surface of *Zip1⁺/⁺* and *Zip1⁻/⁻* MEFs and LLC-GFP-luc cells co-cultured with MEFs (for 1 h in DMEM + 10%FBS + 5 μM TPEN), determined by FACS. LLC-GFP-luc without MEFs were used as a control (Ctl). Data are presented as mean ± SEM, *n* = 3 for each group. Two-tailed *t*-test. Source data are provided as a Source data file (**b**, **d**–**j**). Source data are provided as a Source data file (**a**–**h**).

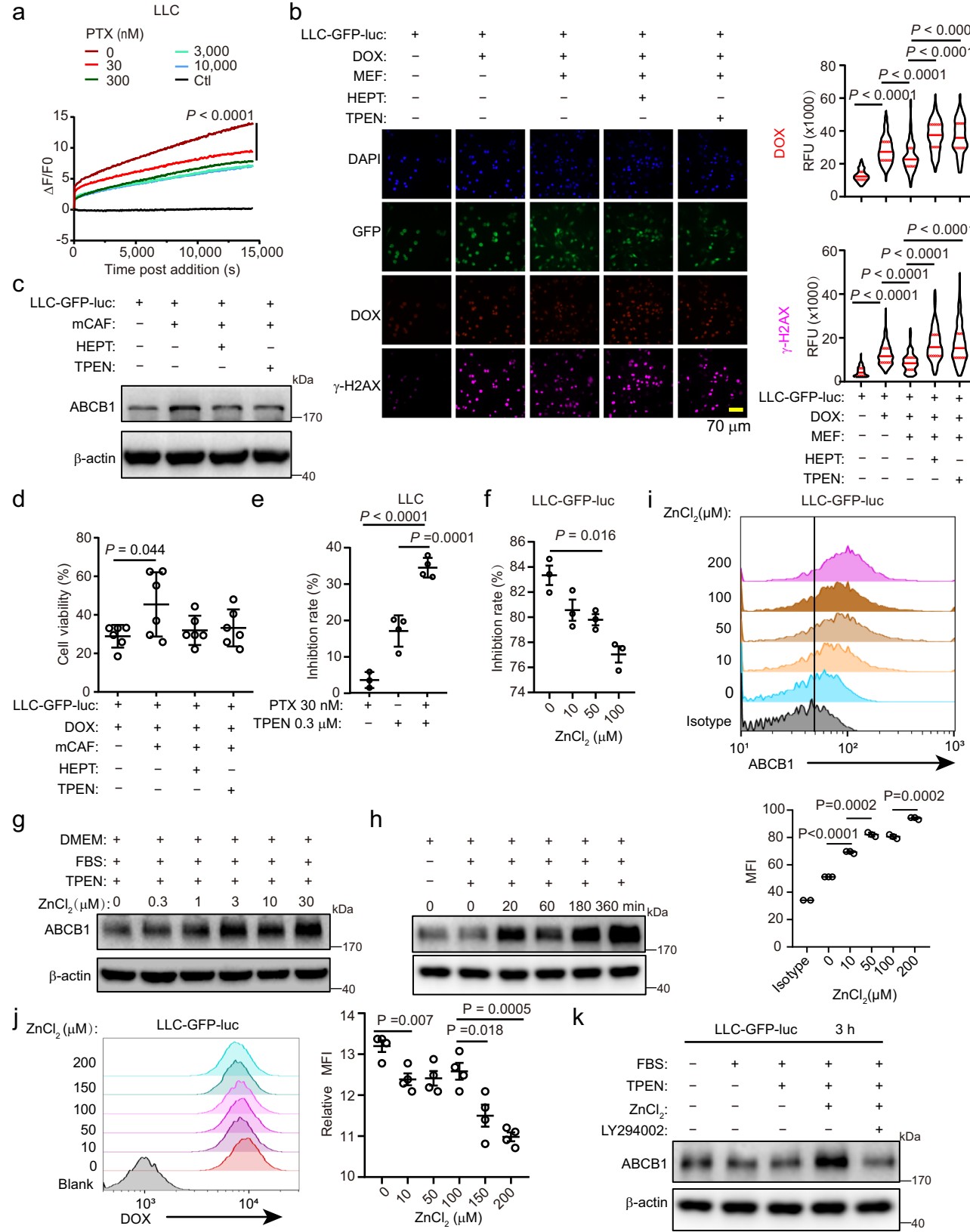

with TPEN reversed these protective effects (Fig. 5b). ABCB1 is a multidrug resistance protein that exports multiple drugs, including DOX, from cells[33], whose inhibition elevated drug accumulation in tumour cells (Supplementary Fig. 5a). Co-culture of LLC-GFP-luc cells with mCAF-induced ABCB1 expression in tumour cells, which was reduced by heptanol or TPEN pretreatment (Fig. 5c, Supplementary Fig. 5b).

When DOX was combined with TPEN or heptanol, the protection of tumour cells by mCAFs was inhibited (Fig. 5d). These results also imply that the resistance of tumour cells to drugs may not be limited to DOX. To examine the resistance of tumour cells to alternative substrates of ABCB1, we combined paclitaxel with TPEN to treat the tumour cells. We added a relatively low level of TPEN (0.3 μM) to reduce tumour

**Fig. 5 | Labile Zn$^{2+}$ induces ABCB1-mediated drug extrusion. a** Zn$^{2+}$ uptake by LLC cells treated with paclitaxel (PTX). ZnCl$_2$ (10 µM) and PTX were added simultaneously. Ctl, no ZnCl$_2$. *n* = 3. **b** DOX accumulation (determined by DOX fluorescence in individual cells) and DNA damage (indicated by γ-H2AX expression in individual cells) in LLC-GFP-luc cells co-cultured with MEFs in DMEM + 10% FBS + TPEN (5 µM) for 4 h. TPEN: 50 µM, pre-treating MEFs for 5 min. HEPT: 2 mM. DOX: 3 µM. Median ± interquartile range. RFU, relative fluorescence unit. **c** ABCB1 expression in LLC-GFP-luc cells co-cultured with mCAFs in DMEM + 10% FBS for 24 h. A representative result from three independent experiments is shown. **d** LLC-GFP-luc cells viability co-cultured with mCAFs under indicated conditions in DMEM + 10% FBS + 5 µM TPEN for 24 h, determined by luciferase activity. DOX: 3 µM. Mean ± SD, *n* = 6. Two-tailed *t*-test. **e** Inhibition rate of PTX on LLC cells with or without TPEN (0.3 µM) in DMEM determined by CCK8. Mean ± SD, *n* = 3 for PTX 30 nM group. *n* = 4 for other groups. One-way ANOVA with multiple comparisons.

**f** Inhibition rate of DOX on LLC-GFP-luc cells with different concentrations of ZnCl$_2$ determined by CCK8. DOX: 3 µM. Cells were treated for 24 h in DMEM + 10% FBS + TPEN (5 µM). Mean ± SEM, *n* = 3. Two-tailed *t*-test. **g, h** ABCB1 expression in LLC-GFP-luc cells treated with indicated concentrations of ZnCl$_2$ for 4 h (**g**), or with 10 µM ZnCl$_2$ for the indicated time (**h**). Representative results from three (**g**) or four (**h**) independent experiments are shown. **i** ABCB1 expression on LLC-GFP-luc tumour cells treated with indicated concentrations of ZnCl$_2$ for 4 h. Mean ± SEM, *n* = 3. Two-tailed *t*-test. **j** DOX accumulation in LLC-GFP-luc cells treated with 3 µM DOX combined with indicated concentrations of ZnCl$_2$ for 4 h. Blank: no DOX treatment. Mean ± SEM, *n* = 4. Two-tailed *t*-test. **k** ABCB1 expression in LLC-GFP-luc cells stimulated with ZnCl$_2$ (30 µM) and LY294002 (50 µM) for 3 h. A representative result from three independent experiments is shown. Two-way ANOVA test for curve comparison. Source data are provided as a Source Data file (**a–k**).

intracellular labile Zn$^{2+}$ without severe tumour cell killing (Fig. 5e). This low level of TPEN dramatically enhanced tumour sensitivity to paclitaxel (Fig. 5e). Moreover, the addition of ZnCl$_2$ significantly attenuated the inhibition of tumour cells by DOX (Fig. 5f) as well as apoptosis induced by DOX (Supplementary Fig. 5c). Consistently, Zn$^{2+}$ stimulated ABCB1 expression and reduced DOX accumulation in tumour cells (Fig. 5g–j, Supplementary Fig. 5d–f). Inhibition of AKT with LY294002 effectively inhibited the upregulation of ABCB1 by Zn$^{2+}$ stimulation in tumour cells (Fig. 5k, Supplementary Fig. 5g), suggesting an AKT-dependent pathway. These results suggest that fibroblast-derived Zn$^{2+}$ induces ABCB1-dependent drug extrusion and protects tumour cells from drug toxicity.

## ZIP1$^+$ fibroblasts promote chemoresistance in mouse lung cancer model

Next, we explored whether ZIP1$^+$ fibroblasts were protective against cancer cells after drug treatment. The results showed that compared with *Zip1$^{-/-}$* MEFs, *Zip1$^{+/+}$* MEFs attenuated DOX-induced DNA damage in LLC-GFP-luc cells treated with DOX for 4 h (Fig. 6a). Correspondingly, *Zip1$^{+/+}$* MEFs reduced DOX and increased the expression of ABCB1 in tumour cells treated with DOX for 4 h (Fig. 6b, c). In addition, *Zip1* knockdown in mCAFs attenuated their protective effects in LLC-GFP-luc tumour cells (Supplementary Fig. 6a). In LLC-GFP-luc tumour-bearing mice, DOX accumulation in tumour cells of *Zip1$^{+/+}$* mice was significantly lower than that in *Zip1$^{+/-}$* and *Zip1$^{-/-}$* mice (Fig. 6d). Consistently, ABCB1 expression was significantly higher in the tumour cells from *Zip1$^{+/+}$* mice than in those from *Zip1$^{-/-}$* mice (Fig. 6e). These results demonstrate that ZIP1 expression in fibroblasts endows lung cancer cells with drug resistance both in vitro and in vivo.

LLC-GFP-luc cells were transplanted into *Zip1$^{+/+}$* and *Zip1$^{-/-}$* mice (Fig. 6f). Compared with *Zip1$^{+/+}$* mice, tumour growth was surprisingly faster in *Zip1$^{-/-}$* mice. When treated with DOX, tumours were more efficiently inhibited in *Zip1$^{-/-}$* mice than in *Zip1$^{+/+}$* mice (Fig. 6f). Next, *Zip1$^{+/+}$* or *Zip1$^{-/-}$* MEFs were co-injected with LLC-GFP-luc tumour cells into WT mice. Compared with *Zip1$^{-/-}$* MEFs, *Zip1$^{+/+}$* fibroblasts promoted tumour growth (Fig.6g), which contrasts with the observation in *Zip1$^{-/-}$* mice, possibly because of the expression of ZIP1 in haematopoietic cells. Upon treatment with DOX, tumours with *Zip1$^{-/-}$* MEFs were significantly inhibited, whereas tumours with *Zip1$^{+/+}$* MEFs were not. Additionally, co-injection of ZIP1-overexpressing PCCAFs did not promote the growth of A549 tumour cells but increased the resistance of these tumour cells to DOX (Supplementary Fig. 6b). Clinical lung cancer commonly has distinct oncogenic mutations such as p53/EGFR/KRAS[2]. H1299 tumour cells (with p53 deletion) were co-injected with hCAF-mock or hCAF-*ZIP1* into NOD-SCID mice. With paclitaxel treatment, the growth of hCAF-mock H1299 tumours was slower than that with hCAF-*ZIP1*, suggesting that ZIP1 expression in hCAF promotes the resistance of H1299 to paclitaxel (Supplementary Fig. 6c). These results demonstrated that ZIP1 expression in fibroblasts protects lung cancer cells from drug toxicity in vivo.

A tetracycline (Tet)-off expression system was used to control the expression of *Zip1* in mCAFs. In this system, ZIP1 was overexpressed under normal conditions but was repressed when tetracycline was added. Although the shutdown of ZIP1 expression by tetracycline in mCAF did not significantly affect tumour growth (Supplementary Fig. 6d), the overexpression of ZIP1 in mCAF promoted tumour growth compared to mCAF-mock (Fig. 6h). To further validate that ZIP1$^+$ fibroblast-dependent upregulation of the multidrug resistance protein ABCB1 expression in tumour cells contributed to chemoresistance, we used paclitaxel combined with doxycycline to treat the co-injected tumours. When tumours were treated with paclitaxel, repression of *Zip1* by doxycycline was observed in mCAF-sensitised tumours to chemotherapy (Fig. 6h).

## S100A4 increases ZIP1 expression in fibroblasts

We sought to determine how chemotherapy treatment-enriched ZIP1$^+$ fibroblasts. The results showed that DOX treatment in vitro reduced ZIP1 expression in mCAF (Fig. 7a, b, Supplementary Fig. 7a, b). Therefore, we suspected that chemotherapy-induced secretory factors might play a role in the regulation of ZIP1 expression in fibroblasts. Other's and our own previous studies demonstrated that S100A4 often increases in tissues under stress[34–37]. In vitro, we found that DOX treatment increased the release of S100A4 into the supernatants by tumour cells (Fig. 7c, Supplementary Fig. 7c). Compared to the control medium, the culture medium collected from DOX-treated tumour cells upregulated ZIP1 expression in fibroblasts, which could be reversed by anti-S100A4 antibody (Fig. 7d, Supplementary Fig. 7d). S100A4 stimulation upregulated ZIP1 expression in fibroblasts in a dose- and time-dependent manner (Fig. 7e, f, Supplementary Fig. 7e, f). Downstream NF-κB (p65), AKT, and p38 were activated by S100A4 in fibroblasts, and their inhibitors impaired S100A4-mediated upregulation of ZIP1 (Fig. 7g–i, Supplementary Fig. 7g–i). RAGE and TLR4 are receptors for S100A4[37]. The upregulation of ZIP1 by S100A4 was attenuated by both RAGE inhibitor (FPS-ZM1) and TLR4 inhibitor (TLR4-IN-C34) (Fig. 7j, Supplementary Fig. 7j). These results suggest that S100A4 might bind to RAGE/TLR4, activate NF-κB (p65), AKT, and p38, and promote the expression of ZIP1 in fibroblasts (Fig. 7k). Furthermore, we confirmed that extracellular S100A4 levels increased in LLC-GFP-luc tumour tissues after DOX treatment (Fig. 7l). These results suggested that the increase in S100A4 levels after chemotherapy may be responsible for the enrichment of ZIP1$^+$ fibroblasts.

## ZIP1$^{high}$ fibroblasts predict chemoresistance in lung cancer patients

To further explore the presence of ZIP1$^+$ fibroblasts in human lung cancer, we analysed a previously published scRNA-seq dataset of human lung adenocarcinoma[38]. Datasets of eight primary lung adenocarcinomas were downloaded and analysed. A total of 554 fibroblasts were identified and classified into four clusters (CH0−3) (Supplementary Fig. 8a, b, Supplementary Data 3). ZIP1 was highly

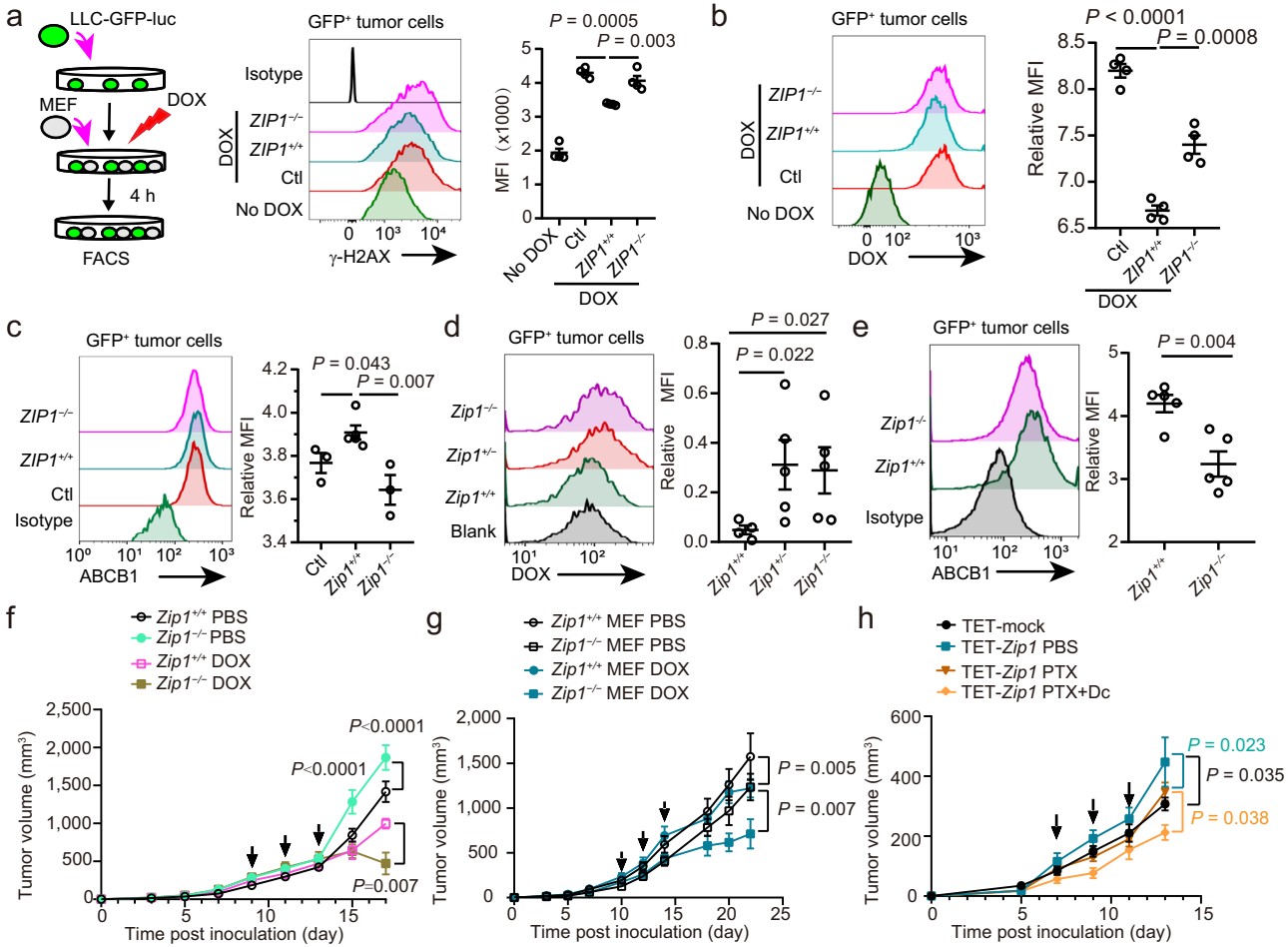

**Fig. 6 | ZIP1⁺ fibroblasts promote chemoresistance of lung cancer cells in vitro and in vivo. a** Experimental design (left) and results of DNA damage (γ-H2AX expression) in LLC-GFP-luc cells co-cultured with *Zip1⁺/⁺* or *Zip1⁻/⁻* MEFs with DOX (3 μM) treatment. Mean ± SEM, *n* = 4. Two-tailed *t*-test. **b** DOX accumulation in LLC-GFP-luc tumour cells co-cultured with *Zip1⁺/⁺* or *Zip1⁻/⁻* MEFs with DOX (3 μM) treatment. Mean ± SEM, *n* = 4. Two-tailed *t*-test. **c** ABCB1 expression in LLC-GFP-luc tumour cells co-cultured with *Zip1⁺/⁺* or *Zip1⁻/⁻* MEFs for 24 h. Mean ± SEM. Ctl, *Zip1⁺/⁺*: *n* = 3; *Zip1⁻/⁻*: *n* = 6. Two-tailed *t*-test. **d** DOX accumulation in LLC-GFP-luc tumour cells 6 h after DOX injection in tumour-bearing *Zip1⁺/⁺*, *Zip1⁺/⁻* and *Zip1⁻/⁻* mice. DOX (10 mg/kg) was intravenously injected into the mice at day 10 post-transplantation. Mean ± SEM. *Zip1⁺/⁺*: *n* = 4; *Zip1⁺/⁻*, *Zip1⁻/⁻*: *n* = 5. One-tailed Kruskal–Wallis test. **e** ABCB1 expression in LLC-GFP-luc cells transplanted into *Zip1⁺/⁺* and *Zip1⁻/⁻* mice at day 10 post-transplantation. Mean ± SEM, *n* = 5. Two-

tailed *t*-test. **f** Tumour growth curves for *Zip1⁺/⁺* and *Zip1⁻/⁻* mice treated with DOX or PBS. Arrows, DOX injection. Mean ± SEM. *P* values are for indicated time points. *Zip1⁺/⁺* PBS (*n* = 11), *Zip1⁻/⁻* PBS (*n* = 12), *Zip1⁺/⁺* DOX (*n* = 8), *Zip1⁻/⁻* MEFs DOX (*n* = 10). **g** Tumour growth curves for co-injection of *Zip1⁺/⁺* or *Zip1⁻/⁻* MEFs with LLC-GFP-luc cells, with DOX or PBS treatment. Arrows indicate DOX injection. *Zip1⁺/⁺*, *Zip1⁻/⁻* MEFs PBS, *Zip1⁻/⁻* MEFs DOX: *n* = 9; *Zip1⁺/⁺* MEFs DOX: *n* = 6. Mean ± SEM. **h** Tumour growth curves for mice inoculated with LLC cells and mCAFs with a tet-off system-controlled expression of *Zip1* (TET-*Zip1*). Mice were administered PTX, with or without doxycycline (Dc) treatment. Mean ± SEM. Arrow, PTX (10 mg/kg) treatment. TET-mock, TET-*Zip1* PBS: *n* = 6; TET-*Zip1* PTX, TET-*Zip1* PTX + Dc: *n* = 5. Two-way ANOVA test for tumour growth curve comparison. Source data are provided as a Source Data file (**a**–**h**).

expressed in the CH0, 1, 3 clusters (Supplementary Fig. 8c, d), among which CH0 was corresponding to a developmental phenotype that enriched developmental pathways including "embryonic organ morphogenesis", "embryonic skeletal system development", "connective tissue development" (Supplementary Data 4). In addition, *CX43* (encoded by *GJA1*) is a marker gene of CH0 fibroblasts, and CH0 fibroblasts expressed S100A4 (Supplementary Data 3, Supplementary Fig. 8e). Correspondingly, the PI3K/AKT signalling pathway was activated in CH0 fibroblasts (Supplementary Fig. 8f). Using the Kaplan–Meier plotter dataset (https://kmplot.com/analysis), we explored the prognostic value of CH0 fibroblasts in patients with cancer receiving chemotherapy. Three marker genes (*ZIP1, S100A4, CX43*) were used as inputs, and overall survival was evaluated. The results showed that high expression of the three marker genes was associated with poor survival in patients with lung cancer receiving chemotherapy (Supplementary Fig. 8g). Interestingly, high expression of the three marker genes was also associated with poor survival in

gastric and ovarian, but not breast, cancer patients with chemotherapy, implying that ZIP1⁺ fibroblasts might contribute to chemoresistance of additional human cancer types.

The clinical relevance of ZIP1⁺ fibroblasts in lung cancer chemoresistance was further investigated in a cohort of 90 lung adenocarcinoma patients who underwent chemotherapy (Supplementary Data 5). First, using immunofluorescence staining, we observed that CX43 was colocalised between ZIP1⁺ fibroblasts and cancer cells in human lung adenocarcinoma, implying gap junction formation between these two cell types (Supplementary Fig. 9a). ZIP1 expression in stromal cells was evaluated, and patients were classified into high or low ZIP1 expression groups according to histological scores. High ZIP1 expression in stromal cells (ZIP1^high) was associated with a poor prognosis (Fig. 8a, b). To evaluate the expression of ZIP1 more specifically in fibroblasts, double-staining of ZIP1 and α-smooth muscle actin (α-SMA) was performed (Fig. 8c). ZIP1 expression in fibroblasts was significantly higher in tumour tissue than in paratumour tissue (Fig. 8d).

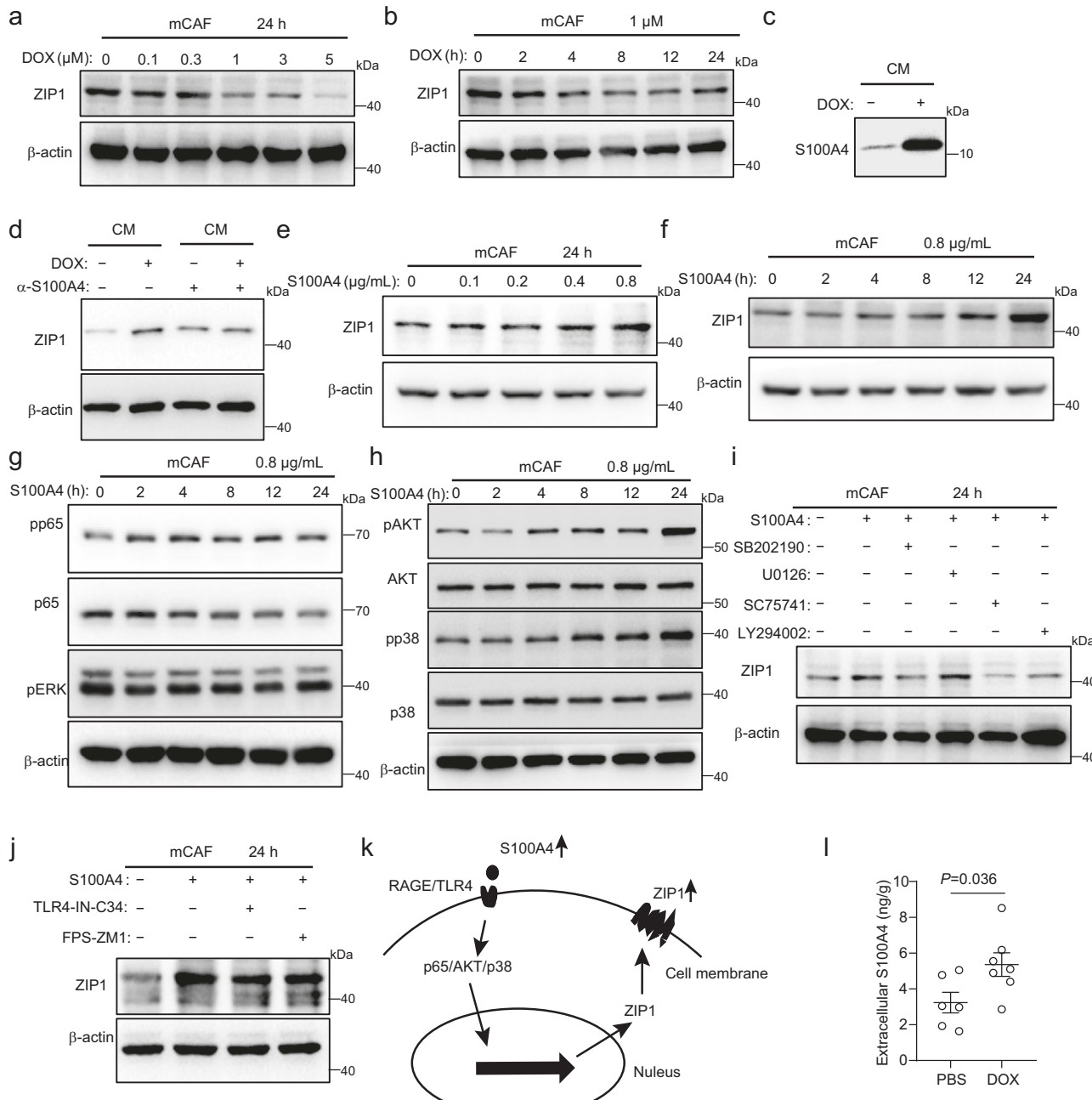

**Fig. 7 | S100A4 increases ZIP1 expression in fibroblasts. a, b** ZIP1 expression in mCAF treated with DOX at the indicated doses (**a**) or for the indicated times (**b**). Representative results from three (**a, b**) independent experiments are shown. **c** S100A4 levels in the same volume of LLC-GFP-luc cell culture mediums (CMs) with or without DOX treatment determined by western blotting. The same number of tumour cells were seeded and pre-treated with DOX (1 μM) for 6 h. Twenty-four hours after changing the fresh medium, CMs were collected. The same volume of CM from different treatment groups was analysed by Western blotting. A representative result from three independent experiments is shown. **d** ZIP1 expression in mCAF treated with indicated CM. For S100A4 neutralisation, CM was pre-incubated with α-S100A4 antibody 3B11 (6 μg/mL) for 1 h. A representative result from four independent experiments is shown. **e, f** ZIP1 expression in mCAF treated with S100A4 at the indicated doses (**e**) or for the indicated times (**f**). β-actin was used as control. The samples derived from the same experiment and blots were processed in parallel. Representative results from three (**e, f**) independent experiments are shown. **g, h** Expression of phosphorylated p65 (pp65), p65, phosphorylated ERK (pERK) (**g**), phosphorylated AKT (pAKT), AKT, phosphorylated p38 (pp38), and p38 (**h**) in mCAF treated with S100A4 for the indicated time periods. β-actin was used as control. The samples derived from the same experiment and blots were processed in parallel. Representative results from three (**g, h**) independent experiments are shown. **i** Expression of ZIP1 in mCAF treated with S100A4, SB202190 (p38 inhibitor, 50 μM), U0126 (ERK inhibitor, 10 μM), SC75741 (p65 inhibitor, 8 μM), and LY294002 (AKT inhibitor, 25 μM) as indicated. A representative result from three independent experiments is shown. **j** Expression of ZIP1 in mCAF treated with S100A4, TLR4-IN-C34 (TLR4 inhibitor, 10 μM), and FPS-ZM1 (RAGE inhibitor, 1 μM), as indicated. A representative result from three independent experiments is shown. **k** Diagram showing the signalling pathways for the regulation of ZIP1 by S100A4. **l** Extracellular S100A4 in LLC-GFP-luc tumours, with or without DOX treatment. *n* = 6 for the PBS group and *n* = 7 for the DOX group. Data are presented as mean ± SEM, *n* = 6 for PBS, *n* = 7 for DOX. Two-tailed *t*-test. Source data are provided as a Source Data file (**a**–**j**, **l**).

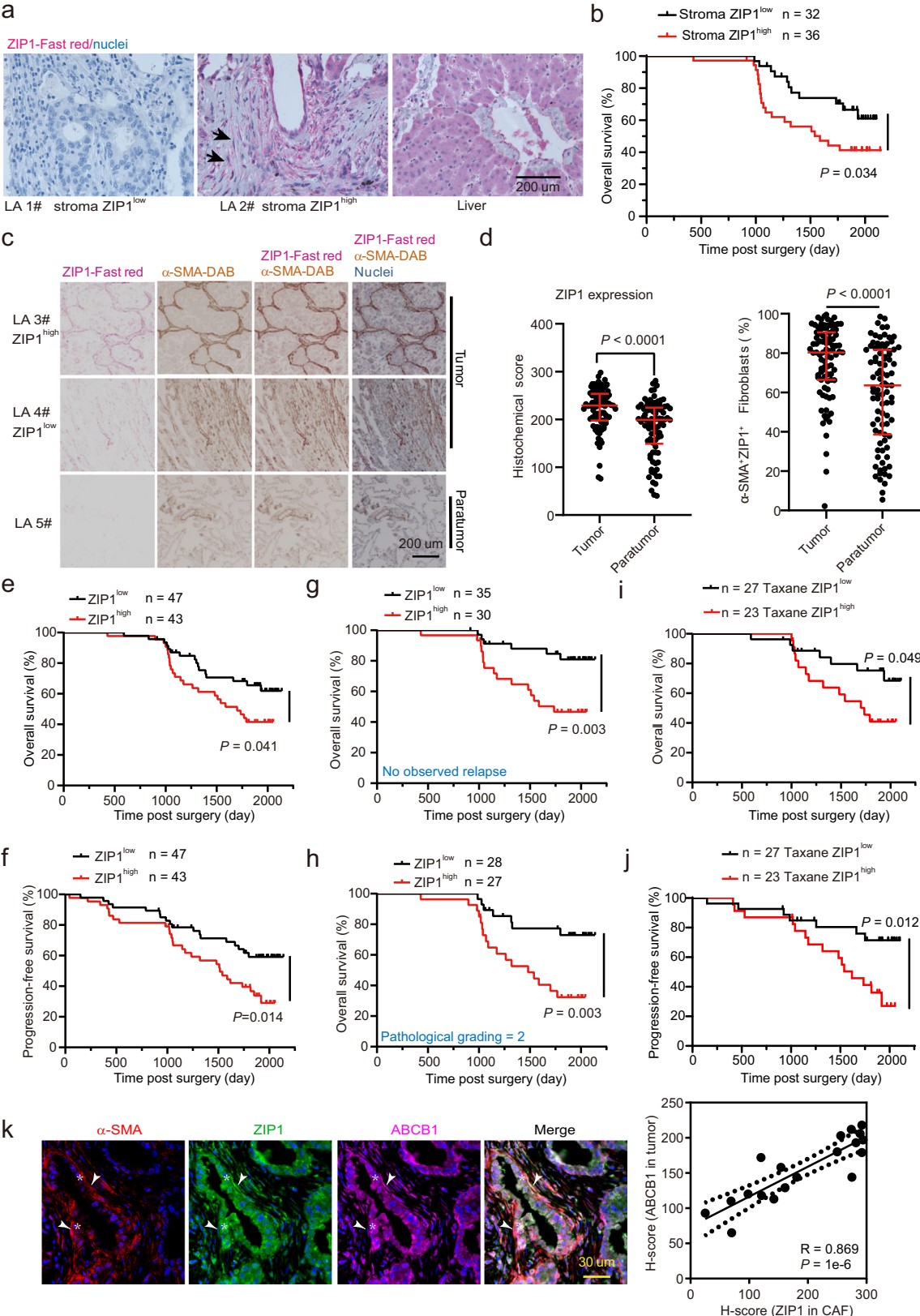

The percentage of α-SMA⁺ZIP1⁺ fibroblasts was also higher in tumour tissues than in paratumour tissues. In most tumours, the proportion of ZIP1⁺ fibroblasts was >50% (Fig. 8d). Moreover, ZIP1^high fibroblast status predicted poor overall and progression-free survival (Fig. 8e, f), indicating an association between ZIP1^high fibroblasts and chemoresistance in lung adenocarcinoma. Although significant associations were

observed between ZIP1 status and mortality and sex, there was no association with pathological grading or clinical stage (Supplementary Fig. 9b). In patients with no observed relapse and those with pathological grade 2, ZIP1^high fibroblast status was an especially powerful predictor of poor survival (Fig. 8g, h). Furthermore, we performed analysis on patients who received paclitaxel or docetaxel (together as

**Fig. 8 | ZIP1^high fibroblasts are associated with poor survival and chemoresistance in patients with lung cancer. a** Representative ZIP1 staining in lung adenocarcinoma (LA) tissues. A human liver section was used as a positive control. Arrow, ZIP1+ fibroblasts. Representative staining results from 68 lung cancer samples are shown. **b** Survival curve of patients with lung adenocarcinoma with stroma ZIP1^high or stroma ZIP1^low status. **c** Double-staining of ZIP1 and α-SMA in lung cancer tissues including tumour and paratumour tissue. Representative staining results from 90 tumour and 90 corresponding paratumour samples are shown. **d** ZIP1 expression levels in fibroblasts and percentage of ZIP1+ fibroblasts in tumour and paratumour tissues (*n* = 90). Data are presented as median with interquartile range.

Two-tailed *t*-test. **e, f** Overall survival (**e**) and progression-free survival (**f**) of patients with lung cancer with ZIP1^high or ZIP1^low status. **g, h** Overall survival of patients with no observed relapse (**g**) or with pathological grade 2 lung adenocarcinoma (**h**). **i, j** Overall survival (**i**) or progression-free survival (**j**) of patients with ZIP1^high and ZIP1^low status in response to chemotherapy with paclitaxel/docetaxel-containing regimens (Taxane). **k** Correlation of ABCB1 expression in tumours with ZIP1 expression in cancer-associated fibroblasts (CAF) (*n* = 19). Representative co-staining of ABCB1, ZIP1 and α-SMA is shown. Arrowhead, ZIP1+ fibroblasts. Asterisk, cancer cell. Log-rank test for survival curve comparison. Mann–Whitney test for two-group comparison. Source data are provided as a Source Data file (**b, d–k**).

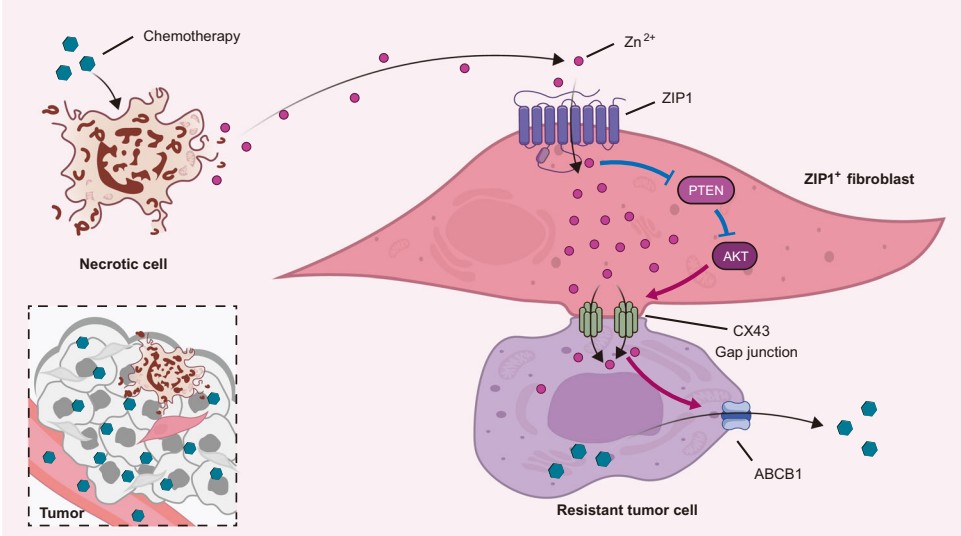

**Fig. 9 | ZIP1+ fibroblasts interconnect lung cancer cells via upregulating CX43 and promote chemoresistance by Zn²⁺ transfer.** Chemotherapy treatment induces Zn²⁺ released from dying cancer cells and the enrichment of ZIP1+ fibroblasts. ZIP1+ fibroblasts interconnect cancer cells with gap junctions by upregulating CX43. Upon treatment, the ability of Zn²⁺ uptake by cancer cells is suppressed.

ZIP1+ fibroblasts act as a reservoir to absorb and transfer Zn²⁺ to lung cancer cells via gap junctions. In ZIP1+ fibroblasts, imported Zn²⁺ stimulates degradation of PTEN and activation of AKT to upregulate the expression of gap junction protein CX43. In lung cancer cells, the elevation of Zn²⁺ enhances the expression of ABCB1 and reduces the accumulation of drugs, resulting in chemoresistance.

"Taxane") (Supplementary Data 5). In these patients, ZIP1^low status was associated with significantly better overall and progression-free survival than ZIP1^high status (Fig. 8i, j), indicating that ZIP1^high status is associated with resistance to taxane-containing regimens. Moreover, in another cohort of 30 lung adenocarcinoma patients (Supplementary Table 4), we found that ABCB1 expression in tumour cells was positively correlated with ZIP1 expression in CAFs (Fig. 8k). These results suggested that ZIP1^high fibroblasts are associated with poor survival and may contribute to chemoresistance in lung cancer. Collectively, our findings revealed that ZIP1+ fibroblasts interconnect lung cancer cells by upregulating CX43 and promoting chemoresistance via Zn²⁺ transfer (Fig. 9).

## Discussion

Short-range fibroblast–tumour interactions, including physical contact during chemotherapy, have scarcely been reported. Early studies reported that fibroblasts coupled with tumour cells through gap junctions transmit bystander cytotoxicity[39]. In contrast, gap junctions between cancer cells and stromal cells may also support tumour growth and chemoresistance[14]. The regulation of fibroblast–tumour gap junction intercellular communication in lung cancer and its effect on chemotherapy are unknown. In this study, we found that *Zip1*+ fibroblasts were enriched in tumours treated with DOX. Unexpectedly, ZIP1+ fibroblasts formed efficient gap junctions with lung cancer cells by upregulating CX43 expression. This coupling enables the transfer of Zn²⁺ from fibroblasts to cancer cells, inducing ABCB1-mediated drug extrusion and chemoresistance. These findings are clinically relevant,

as demonstrated by the association of ZIP1^high fibroblast status with poor survival and chemoresistance in patients with lung cancer. Our results provide insights into short-range fibroblast–tumour interactions and have important implications for the design of strategies to overcome chemoresistance in lung cancer.

ZIP1+ fibroblast enrichment results from adaptation to chemotherapy-induced tissue damage. Chemotherapy kills tumour cells and remodels the tumour stroma. The cellular contents released from dying cells, such as cell debris, HMGB1, ATP, and annexin A1, can be engulfed or sensed by phagocytes as danger-associated molecular patterns (DAMPs) to stimulate long-term adaptive antitumour immunity[40]. In contrast, some cellular contents released from dying cells, such as K+, can generate an immunosuppressive niche by inhibiting T-cell activation[41]. Therefore, cell death may trigger antitumour or protumour responses for unknown reasons. This may reflect diverse cell death types, such as apoptosis or pyroptosis, which usually stimulate rare or strong inflammation[42]. Alternatively, different sensors, such as macrophages or fibroblasts, for released cell contents may exert distinct effects on tumour progression. In this study, we demonstrated that ZIP1+ fibroblasts are enriched after chemotherapy. ZIP1+ fibroblasts probably act as sensors of Zn²⁺ released from dying cells and recycle it for usage, causing tumour drug resistance. Our results suggested that chemotherapeutic drugs do not directly upregulate ZIP1 expression in fibroblasts. Instead, S100A4, which can be induced by chemotherapy, increased the expression of ZIP1 in fibroblasts. In the livers and lungs, we have demonstrated that tissue injury induces an increase in extracellular S100A4[34,35,43]. NF-κB, AKT, and p38

are involved in the S100A4-mediated upregulation of ZIP1. Therefore, we conjecture that chemotherapy-related tissue damage causes an increase in S100A4, which enhances ZIP1 expression in fibroblasts and is possibly responsible for recycling $Zn^{2+}$ in the dying niche.

ZIP1+ fibroblasts uniquely coordinate with tumour cells through gap junctions. Multicellular organisms have evolved diverse strategies to communicate between cells for cellular coordination, including long-range interactions mediated by neural or endocrine, or short-range interactions mediated by direct cell–cell contact[13]. From an organic perspective, tumour and stromal cells adopt similar communication strategies. Although long-range fibroblast–tumour interactions by secretion have been extensively studied, short-range fibroblast–tumour interactions have scarcely been reported[44]. Fibroblasts secrete IL-6 and IL-8 to maintain cancer stem cells[4]. Fibroblasts release glutathione and cysteine to reduce cisplatin accumulation in ovarian cancer[45]. Tumour-derived miRNA-containing exosomes may activate fibroblasts, which in turn can promote lung metastasis[22,46]. Fibroblasts can also modulate tumour vascular function by secreting VEGF[8,47]. For short-range interactions, we and others have reported that fibroblasts engage in physical contact, leading to the invasion of cancer cells[10,11]. In this study, we demonstrated that ZIP1+ fibroblasts specifically upregulate CX43 and form gap junctions with cancer cells. These results suggest that gap junctions possibly enable the direct transfer of $Zn^{2+}$ to tumour cells. Logically, gap junctions may also allow the transmission of other substrates, such as ATP and cGAMP, which requires further study. In support of our results, Maynard et al. reported that cancer cells surviving on therapy progressive disease upregulated the gap junction pathway in lung cancer patients with targeted therapy[48]. ZIP transporters have been reported to transport zinc as well as other metals, such as iron (Fe) and manganese (Mn)[49]. Another interesting question is whether ZIP1+ fibroblasts can transport other metals and affect metal levels in cancer cells. Additionally, ZIP1+ fibroblasts might also have the potential to interact with other stromal cells, such as endothelial and immune cells, to affect tumour progression.

Thus, ZIP1+ fibroblasts may represent a specific fibroblast phenotype. The diverse functions of CAFs may reflect different subtypes of CAFs in tumours[7,8]. An inflammatory CAF (iCAF) phenotype has been identified in different cancer types which recruit macrophages to establish an immunosuppressive microenvironment[7]. In lung cancer, scRNA-seq has previously identified five clusters of fibroblasts, including two ECM phenotypes (ecmCAFs), one myCAF, and two metabolic phenotypes[50]. In this study, fibroblast Cluster 6 was consistent with myCAFs and Cluster 7 resembled ecmCAFs. ZIP1+ fibroblasts are enriched after chemotherapy and have enhanced abilities to form gap junctions and absorb and transfer $Zn^{2+}$ to lung cancer cells, leading to chemoresistance. Therefore, we argue that ZIP1+ fibroblasts represent a CAF subtype with a developmental phenotype and zinc transport activity. In both mouse and human lung cancers, these fibroblasts are *ZIP1+S100A4+CX43*high CAF. We propose that *ZIP1+S100A4+CX43*high CAF may be called zinc-transport CAF (zCAF), which absorbs and transfers $Zn^{2+}$ to neighbouring cancer cells through gap junctions. The PI3K/AKT signalling pathway probably drives the generation of zCAF. Except for zCAF, our results indicate that ZIP1 might also be expressed in fibroblasts with low gap junction protein expression, which may reflect low levels of $Zn^{2+}$ in these cells, and these cells might have the potential to become zCAF when encountering adequate $Zn^{2+}$. To further define the specific function of zCAF, it will be useful to develop strategies for the isolation of zCAFs in future studies.

$Zn^{2+}$ transfer by ZIP1+ fibroblasts contributes to tumour chemoresistance. Zinc is an essential trace element for cell proliferation and survival[15]. Besides structural $Zn^{2+}$, there is a small amount of loosely bound $Zn^{2+}$, known as free or labile $Zn^{2+}$, representing a physiological source of accessible $Zn^{2+}$[51]. Reduced cytoplasmic $Zn^{2+}$ related to ZIP7 deficiency leads to B cell development failure[21]. ZIP14-mediated $Zn^{2+}$ uptake in muscle progenitor cells represses the expression of MyoD and Mef2c and blocks muscle cell differentiation, driving cancer-induced cachexia[18]. $Zn^{2+}$ imported by ZIP4 has been linked to cancer cell proliferation, resistance to apoptosis, cachexia in pancreatic cancer[19], and metastasis in lung cancer[52]. It has been reported that serum $Zn^{2+}$ levels are low in many cancers, including lung cancer[53,54]. However, $Zn^{2+}$ dynamics are unclear, especially in tumour chemotherapy. We found that chemotherapy-induced cancer cell necrosis, releasing labile $Zn^{2+}$ into the extracellular space. Upon treatment, cancer cells were suppressed to take up $Zn^{2+}$ from the extracellular space, possibly resulting in zinc-deficient conditions for cancer cells. ZIP1+ fibroblasts can act as $Zn^{2+}$ reservoirs to absorb $Zn^{2+}$ and transfer it to cancer cells through gap junctions, thereby leading to ABCB1-mediated drug extrusion in lung cancer cells. Brain endothelial cells use a similar mechanism to prevent drug toxicity[55]. Our findings demonstrate that ZIP1-mediated upregulation of CX43 is fibroblast-origin independent, implying that this mechanism might apply to both primary tumours and distant metastases of lung cancer treated with chemotherapy. In fibroblasts, $Zn^{2+}$ stimulated PTEN degradation, consistent with previous findings[56]. Inhibition of PTEN by $Zn^{2+}$ has been linked to AKT activation and CX43 upregulation, paving the way for $Zn^{2+}$ transfer. A study has reported that $Zn^{2+}$ reduced CX43 expression in astrocytes[57], possibly because of the different cell types. In fibroblasts with zinc transport deficiencies, vimentin-filament stability is impaired[58]. Thus, $Zn^{2+}$ in fibroblasts and cancer cells may also regulate other signalling pathways that require further exploration.

ZIP1+ fibroblasts are clinically relevant in patients with lung cancer. We found that expression of ZIP1 in the tumour stroma, specifically in fibroblasts, was associated with chemotherapy resistance in lung adenocarcinoma, and correlated with ABCB1 expression in the tumour cells of lung cancer patients. Specifically, ZIP1high fibroblast status was associated with resistance to taxane-containing drug regimens. Therefore, clinical testing of fibroblast ZIP1 expression in lung cancer could inform the choice of therapeutic drug regimens. Considering that a series of therapeutic drugs are substrates of ABCB1, the association between ZIP1high fibroblasts and multiple chemotherapies merits further study. Finally, our results imply that ZIP1+ fibroblasts may also contribute to chemoresistance in other cancer types, such as gastric cancer, which requires further study. Taken together, our results reveal a special fibroblast–lung cancer cell short-range interaction via CX43-mediated gap junctions, which contributes to chemoresistance via $Zn^{2+}$ transfer.

## Methods

### Animals
The Ethics Committee of Scientific Research and Clinical Trial at the First Affiliated Hospital of Zhengzhou University approved this study (2020-KY-349, 2021-KY-0626-002). C57BL/6-EGFP mice were originally from Osaka University (Osaka, Japan)[59]. Female C57BL/6, nude, and NOD-SCID mice were purchased from Vital River Laboratories (VRL, Beijing, China). ZIP1-deficient mice (*Slc39a1*−) on a C57BL/6 background were purchased from Cyagen (KOCMP-30791-Slc39a1-B6N; Suzhou, China). Exon 3 of the *Slc39a1/Zip1* gene was deleted by CRISPR/Cas-mediated genome engineering. *Zip1+/+*, *Zip1+/−* and *Zip1−/−* mice were identified using two pairs of PCR primers (Primer pair 1: F1: 5′-TCTCTGTATAGGTCCAAGAGTCAC-3′, R1: 5′-TGAAGGAGGGACAAG CAAATA. Primer pair 2: F2: 5′-CATGTGACGGTAAGCATTGACTTC-3′, R1: 5′-TGAAGGAGGGACAAGCAAATAAGTG-3′). Female mice aged 6–8 weeks were used. All mice were housed at 21 °C and 50% humidity, on average, on a 12 h light/dark cycle in a specific pathogen-free facility at Zhengzhou University.

### Cell culture
The mouse lung cancer cell line LLC and the human lung cancer cell lines A549 and H1299 (ATCC; LGC Standards) were cultured in

Dulbecco's modified Eagle's medium (DMEM; HyClone, UT, USA) supplemented with 10% foetal bovine serum (FBS) (PAN biotech, Bavaria, Germany) and 100 IU/mL penicillin/streptomycin (Gibco, MA, USA). Cells were incubated at 37 °C in a humidified atmosphere of 5% $CO_2$ and 95% air and checked monthly for mycoplasma contamination (MP0035; Sigma-Aldrich, Shanghai, China). In the indicated experiments, 5 μM TPEN was added to the culture medium to neutralise $Zn^{2+}$ in the FBS. The stable LLC-GFP-luc cell line was constructed by transfecting LLC cells with a GFP-luc lentivirus vector (Genechem, Shanghai, China). PCCAFs were kindly provided by Prof. Ju Zhang at the College of Life Sciences, Nankai University, and were cultured in DMEM supplemented with 10% FBS[60].

The mCAFs were LLC tumour-associated fibroblasts that were isolated as previously described[11]. Briefly, tissues of transplanted LLC tumours (~8 mm in diameter) were separated and digested in DMEM containing 0.1 mg/mL collagenase IV (Sigma, C5138) for 2 h at 37 °C. Cell suspensions were cultured, and on the following day, the suspended tumour cells were removed by washing the cell layers with fresh medium. The remaining adherent fibroblasts were passaged and identified by immunostaining for the fibroblast markers α-SMA (1:100, ab21027, Abcam, UK) and vimentin (1:100, GTX100619, GeneTex, USA).

MEFs were isolated from mice as previously described[59]. Briefly, pregnant females were sacrificed on day E14, and each embryo was separated from the placenta. After discarding the brain and other dark red organs, the embryos were separately minced and suspended in 0.1 mg/mL collagenase IV (Sigma-Aldrich, C5138) and incubated at 37 °C for 30 min. Suspended cells were plated in DMEM. The medium was changed the following day, leaving behind the adherent MEFs. In this study, ZIP-deficient MEFs and control cells were isolated from the embryos of female ZIP1$^{+/-}$ mice mated with ZIP1$^{+/-}$ mice. The genotypes of the MEFs were determined by PCR.

### Isolation of human lung cancer-associated fibroblasts
Primary hCAFs were isolated from a patient with lung adenocarcinoma (EGFR exon 19 deletion) following surgical resection with neoadjuvant chemotherapy at the First Affiliated Hospital of Zhengzhou University, Henan, China. Written informed consent was obtained from all patients. The study was approved by the Ethics Committee of Scientific Research and Clinical Trial at the First Affiliated Hospital of Zhengzhou University (2020-KY-349). Briefly, the tumour tissues were minced, suspended in 1 mg/mL collagenase IV (Sigma-Aldrich, C5138), and incubated at 37 °C for 60 min. Suspended cells were plated in DMEM. The medium was changed the following day, leaving behind an adherent mix of tumour cells and fibroblasts. After expansion, the fibroblasts and cancer cells were separated using human anti-fibroblast beads (130-050-601, Milteny, Shanghai, China). Purified fibroblasts were used as the hCAFs.

### Tumour transplantation and treatment
Tumour transplantation was performed as previously described[47]. Except for indicated, tumour cells ($1 \times 10^6$) were injected subcutaneously into mice. Tumour growth was monitored every 2–3 days. Tumour volumes were calculated as length × width$^2$ × 0.5. All tumour-bearing mice were humanely euthanized prior to their tumours reaching the maximally allowed tumour size (20 mm in diameter) in our animal protocol. When MEFs or PCCAFs were used as stromal fibroblasts, LLC-GFP-luc or A549 tumour cells ($3 \times 10^5$) were mixed with fibroblasts ($9 \times 10^5$) in 100 μL PBS and then subcutaneously injected into naïve mice. Doxorubicin (DOX, 5 mg/kg) or paclitaxel (PTX, 10 mg/kg) was administered intraperitoneally three times every 2 days when the tumour volume reached ~100 mm$^3$.

### Stromal cell sorting for single-cell RNA sequencing
To prepare cell suspensions for scRNA-seq, LLC tumour cells ($1 \times 10^6$) were injected subcutaneously into eight C57BL/6-EGFP mice. Four

mice were treated with DOX three times every 2 days. The control group was treated with PBS. Two days after the last DOX treatment, the tumour tissue was collected and digested with 1 mg/mL collagenase IV at 37 °C for 60 min. The cell suspension was washed with PBS + 2 mM EDTA + 1% BSA and stained with CD45-AF700 (1:100, BioLegend). To exclude dead cells, the cells were re-suspended in 0.5 μg/mL propidium iodide before analysis. FACS analysis and cell sorting were performed using a FACS Aria (Becton Dickinson, NY, USA). EGFP$^+$CD45$^-$ stromal cells from three mice per group were sorted and mixed for scRNA-seq.

### Cell capture and cDNA synthesis
Using the Chromium Single Cell 3′ GEM Library & Gel Bead Kit v3 (10x Genomics, Shanghai, China, 1000075) and Chromium Single Cell B Chip Kit (10x Genomics, 1000074), the cell suspension (1000 living cells per microlitre determined by Count Star) was loaded onto the Chromium single-cell controller (10x Genomics) to generate single-cell gel beads in the emulsion, according to the manufacturer's protocol. Briefly, single cells were suspended in PBS containing 0.04% BSA. ~12,000 cells were added to each channel and target cell recovery was estimated to be ~6000 cells. Captured cells were lysed and the released RNAs were barcoded through reverse transcription in individual GEMs. Reverse transcription was performed on an S1000TM Touch Thermal Cycler (Bio Rad) at 53 °C for 45 min, followed by 85 °C for 5 min, and held at 4 °C. The cDNA was generated and amplified, and the quality was assessed using an Agilent 4200 (CapitalBio Technology, Beijing, China).

### scRNAseq library preparation
According to the manufacturer's instructions, scRNA-seq libraries were constructed using Chromium Single Cell 3′ GEM Library & Gel Bead Kit v3. The libraries were sequenced using an Illumina Novaseq6000 sequencer with a sequencing depth of at least 100,000 reads per cell using a paired-end 150 bp (PE150) reading strategy (performed by CapitalBio Technology).

The raw sequence data of mouse EGFP$^+$CD45$^-$ stromal cell scRNA-seq reported in this paper have been deposited in the Genome Sequence Archive[61] of the National Genomics Data Center[62], China National Center for Bioinformation/Beijing Institute of Genomics, Chinese Academy of Sciences, under accession number CRA004556.

Single-cell sequencing datasets of human lung adenocarcinoma were downloaded from NCBI's Gene Expression Omnibus (GSE123904), which has been previously published[38].

### Alignment and quantification
Cell Ranger v2.1.0 with default settings was used to perform read alignment, barcode processing, and single-cell gene counting. Briefly, the FASTQ files for each library were processed individually using the Cell Ranger Count pipeline, which made use of STAR v.2.5.1b (Alexander Dobin) to align cDNA reads to the mm10 mouse reference genome. The cell barcodes and UMI associated with the aligned reads were filtered and corrected. The retained reads were used to collapse the PCR duplicates and accurately quantify the number of transcript molecules captured for each gene in each cell.

### Quality control and batch-effect correction
Seurat v2.1.0 (R version 3.6.0, the R project) was used to control cell and gene quality. Cells containing fewer than 1000 expressed genes were removed. Genes that were expressed in fewer than 10 cells were also removed. For each cell, the counts were normalised to the total counts and then multiplied by a scale factor of 100,000 before transforming to the log2 scale. To identify highly variable genes, the relationship between mean expression and dispersion was fitted using LogVMR as a dispersion function. FindIntegrationAnchors and IntegrateData functions with default parameters were used to correct the

batch effect and merge the datasets. The count matrix is then scaled, centred, and used for dimensionality reduction and clustering.

## Dimensionality reduction, clustering and marker selection

To reduce the dimensionality of the dataset, the variably expressed genes were summarised using principal component analysis, and the first 20 principal components were summarised using UMAP dimensionality reduction using the default settings of the RunUMAP function. The scores of cells along these principal components were used to build a *k*-nearest-neighbour graph and partition the cells into transcriptionally distinct clusters using the smart local moving community detection algorithm, as implemented in the FindClusters function in Seurat. To identify marker genes for each cluster, we contrasted cells from that cluster with all other cells of that cluster using the Seurat FindMarkers function. Marker genes were required to have an AUC value > 0.7 using ROC analysis to classify cells between clusters. Additionally, marker genes were required to have the highest mean expression in that cluster out of all clusters.

## Gene-set variation analysis (GSVA)

Pathway analyses were predominantly performed on the 50 hallmark pathways described in the molecular signature database and exported using the msigdbr package v7.4.1. GSVA was applied using standard settings as implemented in the GSVA package v1.40.1. The pathway activity scores of the cells were used to calculate *t*-values using the moderated *t*-test in Limma v.3.36.2. The *t*-values were plotted as a heatmap using ggplot2 v.2.2.1.

## RNA velocity

For RNA velocity analysis, the spliced and unspliced matrices of reads were summarised using velocyto v.0.17.17[63] with default parameters from aligned bam files generated by CellRanger. The resulting loom object for each sample was loaded and processed using velocyto.R v0.6. Latent time and putative driver genes were calculated using the dynamical velocity model from scVelo v.0.2.4[64].

## Gene expression programmes (GEPs) analysis

The cNMF algorithm v1.4 (Dylan Kotliar)[28] was applied to the cells of clusters 0–3 to identify activation programmes (GEPs). In brief, non-negative matrix factorisation was run 100 times for *k* from 4 to 10 signatures. A *k* of 6 was chosen with *k*_selection_plot. An outlier threshold of 0.1 was selected to complete the consensus step for cNMF.

## Pathway activity estimation

To compare pathway activity among seven clusters, we used PROGENy v1.14.0 to estimate the activity of 14 pathways using the top 500 most responsive genes from the model as it is recommended from a benchmark study[29].

## RNA extraction and qPCR

To detect the mRNA expression of genes, total RNA was extracted from the samples using TRIzol reagent (9109, Takara, Shiga, Japan) according to the manufacturer's instructions. First-strand cDNA was synthesised from 1 μg of total RNA using a Prime Script RT reagent kit (RR047A, Takara). qPCR was performed using SYBR Premix Ex Taq II (RR820A; Takara) and assessed using an Agilent Mx3005P instrument. The abundance of mRNA for each gene of interest was normalised to that of 18S rRNA. The following primers were used: mZip1 Fw, GCTTCGAA GGTCAGGTGCTA; mZip1 Rv, GCAGCAGGTCCAGAAGACAT; hZIP1 Fw, TGAGCCTAGTAAGCTGTTTCGC; hZIP1 Rv, CAGGGCCTCATCTATGG CA; 18S, ACCGCAGCTAGGAATAATGGA; CAAATGCTTTCGCTCTGGTC.

## Transfection

For transfection, fibroblasts were seeded at a density of $2 \times 10^5$ cells/well in six-well plates. The following day, DNA constructs or siRNA

targeting *Zip1* and the control were mixed with lipofectamine-3000 (Lipo, Invitrogen, Shanghai, China) in Opti-MEM (1:1, v/v), and the mixture was incubated for 15 min at room temperature. The DNA–lipo complexes were then added to the cells. For knockdown, experiments were performed 2 days after transfection. For overexpression, the transfected cells were screened for 7–10 days by the addition of corresponding antibiotics. After selection, the antibiotics were removed and the cells were used for experiments. pcDNA3.1-mZip1, carrying mouse *Zip1* cDNA under the control of the CMV promoter, was constructed by Sangon Biotech (Shanghai, China). pCMV-hZIP1 carrying human *ZIP1* cDNA under the control of the CMV promoter was purchased from Sino Biological (HG16650-NM; Beijing, China). Tetracycline (Tet)-off-system-controlled *Zip1* expression was constructed as previously described[65]. The mCAFs were first transfected with pTet-off (BD Biosciences, San Jose, CA, USA). After selection, cells were transfected with pTRE2-mZip1 carrying mouse *Zip1* cDNA under the control of a mutant CMV promoter (TET-*Zip1*) or empty pTRE2 (TET-mock).

## Western-blotting analysis

Cells were lysed with RIPA lysis buffer and the lysates were collected. Proteins were separated by 10% SDS–PAGE and transferred onto polyvinylidene difluoride (PVDF) membranes. The membrane was incubated with the primary antibody, and the specific binding of the primary antibody was detected with HRP-conjugated goat anti-rabbit (1:2000, 31460) or goat anti-mouse (1:2000, 31430) secondary antibody (Invitrogen). The following primary antibodies were used: ZIP1 (1:1000, AL0-AZT-001; Alomone, Israel), AKT (1:1000, clone 40D4, 2920; CST, USA), pAKT (1:1000, clone D25E6, 13038; CST, USA), PTEN (1:1000, clone D4.3, 5832; CST, USA), pp38 (1:1000, clone D3F9, 4511; CST, USA), p38 (1:1000, clone D13E1, 8690; CST, USA), pp65 (1:1000, clone 93H1, 3033; CST, USA), p65 (1:1000, clone D14E12, 8242; CST, USA), pERK (1:1000, clone D13.14.4E, 4370; CST, USA), ABCB1 (1:1000, clone C219, MA1-26528; Invitrogen, USA), GFP (1:1000, clone GF28R, MA5-15256, Invitrogen). RFP (1:1000, clone RF5R, MA5-15257 Invitrogen), CX43 (1:1000, 26980-1-AP; Proteintech Group, China), S100A4 (1:1000, clone 3B11, kept in our Lab) and β-actin (1:6000, clone 2D4H5, 66009-1-Ig; Proteintech, Wuhan, China).

## DNA damage and DOX content

$5 \times 10^4$ LLC-GFP-luc cells were seeded into individual wells of 12-well plates and cultured overnight. The next day, $1 \times 10^5$ $Zip1^{+/+}$ or $Zip1^{-/-}$ MEFs were added to the culture in DMEM + 10% FBS + 5 μM TPEN. DOX (3 μM) was then added. After 4 h, to detect DNA damage, cells were collected and stained with GFP (1:100, clone B-2, AG281, Beyotime, Shanghai, China) and γ-H2AX (1:200, clone 20E3, 9718, CST, USA). Donkey anti-mouse (1:200, A-21202) and anti-rabbit AF647 (1:200, A-31573, Invitrogen, Shanghai, China) secondary antibodies were used to detect GFP and γ-H2AX, respectively. The expression of γ-H2AX in individual tumour cells was analysed using FACS or confocal microscopy. To detect DOX content in individual tumour cells, the cells were collected, and DOX fluorescence was directly determined by FACS at FL3 or confocal microscopy. To detect ABCB1 or CX43 expression, The cells were stained with appropriate antibodies. After washing, the cells were stained with donkey anti-rabbit AF647 antibody (1:200, A-31573) and analysed using FACS. To quantify the fluorescence, the mean fluorescence intensity (MFI) or relative MFI was calculated. Relative MFI was calculated as the MFI of the sample/MFI of the negative control-1.

## Flow cytometry

Rabbit anti-ABCB1 (1:100, 22336-1-AP, Proteintech) and rabbit anti-CX43 (1:100, 26980-1-AP, Proteintech) were used as primary antibodies to stain the cell suspensions, and Alexa Fluor 647 donkey anti-rabbit IgG (H + L) (1:200, A-31573, Thermo Fisher Scientific) was used as the secondary antibody. PE/Cy7-anti-mouse CD31 (1:100, 102418,

BioLegend) and Alexa Fluor 700 anti-mouse CD45 antibodies (1:100, 147716, BioLegend) were used to stain the endothelial and immune cells. Dead cells in samples were excluded by propidium iodide (PI) staining in indicated experiments. Samples were analysed using Canto II with BD FACSDiva software v8.0.1, and the results were analysed using FlowJo v10 (BD Bioscience, USA). Gating strategies have been provided in the figures as well as in Supplementary Fig. 10.

## Histopathology and immunostaining
The Ethics Committee of Scientific Research and Clinical Trial at the First Affiliated Hospital of Zhengzhou University approved this study (2020-KY-349, 2021-KY-0626-002). Tumour tissues were harvested and embedded in the O.C.T. compound, frozen, and cut using a cryostat. For histological staining of tumour tissues, frozen sections (7 μm thick) were stained with rat anti-ER-TR7 (1:100, clone ER-TR7, ab51824, Abcam, Cambridge, UK), goat anti-α-SMA (1:100, ab21027, Abcam), and rabbit anti-ZIP1 (1:100, AL0-AZT-001, Alomone, Israel) antibodies. A human lung adenocarcinoma tissue microarray (TMA) with survival data was purchased from Zuocheng Bio (TLungAde-01, Shanghai, China). The TMA represented 90 lung adenocarcinoma cases that were treated with chemotherapy. Another TMA, representing 30 lung adenocarcinoma cases, was purchased from Outdo Biotech (HLugA060PG02, Shanghai, China). For immunohistochemistry, tissue sections were deparaffinised and incubated in citrate buffer at 95 °C for 40 min for antigen retrieval, and then incubated overnight at 4 °C with primary antibodies anti-ZIP1 (1:100 dilution) and anti-α-SMA (1:100 dilution). After three washes, the tissue sections were incubated with AP-conjugated anti-rabbit antibody and HRP-conjugated anti-goat antibody (1:200 dilution) for 1 h at room temperature. After three washes with PBS, Fast Red or DAB solution was added and the slides were counterstained with haematoxylin. Negative controls were treated in the same manner, except without the addition of primary antibodies. Co-staining of ZIP1, α-SMA, and ABCB1 in TMAs was performed according to the manufacturer's instructions using a PANO multiplex IHC 6-colour kit (PN100718AD, PANOVUE, Beijing, China). ZIP1 was labelled with PPD620, α-SMA with PPD520, and ABCB1 with 540. The results of immunohistochemical staining were acquired and assessed a Vectra 3.0 (Perkin Elmer). The tumour and stroma areas were auto-recognised by training the machine according to staining with Inform software 2.1.1. The percentage of immunoreactive cells and staining intensity were evaluated using the Inform software. The staining extent score was on a scale of 0–300, corresponding to the percentage of immunoreactive stromal cells (0–10%, 11–25%, 26–75%, and 76–100%) and staining intensity (negative, score = 0; weak, score = 1; strong, score = 2; extraordinarily strong, score = 3). A score ranging from 0 to 300 was calculated using the staining extent score × intensity score × 100. The cutoff of the histological score for ZIP1 single-staining was 150 (low, 0–150; high, 150–300) and for ZIP1/α-SMA double-staining was 230 (low, 0–230; high, 230–300). The expression level of ZIP1 in each specimen was defined as low or high.

## Inhibition rate and cell viability
Fibroblasts or tumour cells ($5 \times 10^3$) were seeded into individual wells of 96-well plates. The next day, DOX and the drugs were added to treat the cells for 24 h, and the remaining live cells were detected using a CCK8 kit. Untreated cells were used as the controls. The inhibition rate was calculated as (OD450[control]− OD450[sample])/OD450[control]×100%.

To evaluate the protective effect of fibroblasts on tumour cell viability, MEFs were co-cultured with LLC-GFP-luc cells and treated with DOX. Briefly, LLC-GFP-luc cells ($5 \times 10^3$) were seeded into 96-well plates and cultured overnight. The following day, MEFs ($1 \times 10^4$) were added to the cultures. DOX (3 μM) was added to the cells. Alternatively, MEFs ($1 \times 10^4$) were seeded into the plate first, and the next day LLC-GFP-luc cells ($1 \times 10^4$) were added. DOX (1 μM) was added as a treatment. After 24 h, 1.5 mg/mL D-luciferin (40801ES03, YEASON, Shanghai, China) was added to the culture to detect luciferase activity in LLC-GFP-luc cells. Untreated LLC-GFP-luc cells were used as controls. Tumour cell viability was evaluated by the activity of luciferase in LLC-GFP-luc cells as luminescence [sample]/luminescence [control]×100%.

## Intracellular $Zn^{2+}$ determination by FACS
To detect intracellular $Zn^{2+}$ using FACS, $5 \times 10^4$ LLC cells were seeded into individual wells of 12-well plates and cultured overnight. The following day, LLC cells were loaded with 2 μM FluoZin3-AM (F24195, Invitrogen) for 1 h at room temperature and treated with different concentrations of $ZnCl_2$ in DMEM + 10% FBS + 5 μM TPEN (P4413, Sigma-Aldrich) for 20 min. After treatment, the cells were collected and FluoZin3 fluorescence was analysed using Canto II (Becton Dickinson). LLC cells treated with FluoZin3 were treated with 5 μM TPEN in FACS buffer and used as a negative control. For the co-culture of A549 and hCAFs, A549 cells were pre-stained with 3 μM CM-Dil (40718ES60, Yeason) at 37 °C for 30 min in DMEM. hCAFs were then added to the culture in DMEM supplemented with 10% FBS. After 3 h of co-culture, the cells were loaded with 2 μM FluoZin3-AM at room temperature for 1 h in DMEM and analysed using FACS. Cells without FluoZin3 labelling were used as negative controls.

## $Zn^{2+}$ transfer
$5 \times 10^3$ A549 cells were seeded into individual wells of 96-well black plates (165305, NUNC, Shanghai, China) and cultured overnight. After loading with 2 μM FluoZin3-AM for 1 h, the A549 cells were co-cultured with $1 \times 10^4$ hCAFs for 2 h in DMEM. After changing DMEM to HBSS (136.9 mM NaCl, 5.4 mM KCl, 0.4 mM $KH_2PO_4$, 0.3 mM $Na_2HPO_4$, 25 mM HEPES, 5.6 mM D-glucose) supplemented with 1.26 mM $CaCl_2$, the FluoZin3 fluorescence was read by SpectraMAX i3X (Molecular Devices, CA, USA) at 525 nm to evaluate the intracellular labile $Zn^{2+}$.

Alternatively, $5 \times 10^3$ LLC or A549 cells were seeded into individual wells of 96-well black plates and cultured overnight. After loading with 2 μM FluoZin3-AM for 1 h, LLC or A549 cell culture was added with $1 \times 10^4$ MEFs or hCAFs in HBSS (+1.26 mM $CaCl_2$). Sixty seconds after addition, FluoZin3 fluorescence was read every 30 s for the indicated times to evaluate the changes in intracellular labile $Zn^{2+}$ in tumour cells. All values minus the values of the blank group were used to evaluate intracellular $Zn^{2+}$. Six-time points of tumour-cell fluorescence ($F$) without co-culture at the beginning were used as $F0$, and the alteration in intracellular $Zn^{2+}$ was calculated as $\Delta F/F0$, where $\Delta F = F-F0$. The origin was forced to zero, and 60 s was applied to the start time of the measurement.

## $Zn^{2+}$ uptake
To detect $Zn^{2+}$ uptake, $1 \times 10^4$ fibroblasts or tumour cells were seeded into individual wells of 96-well black plates and cultured overnight. After loading with 2 μM FluoZin3-AM for 1 h, 30 μM $ZnCl_2$ was added to HBSS. Sixty seconds after addition, FluoZin3 fluorescence was read every 30 s for the indicated times to evaluate the changes in intracellular labile $Zn^{2+}$ in cells. The TPEN treatment was used as the negative control. All values minus the values of the blank group were used to evaluate intracellular $Zn^{2+}$. Six-time points of cell fluorescence ($F$) without co-culture at the beginning were used as $F0$, and the alteration in intracellular $Zn^{2+}$ was calculated as $\Delta F/F0$, where $\Delta F = F-F0$. The origin was forced to zero, and 60 s was applied to the start time of the measurement.

## $Zn^{2+}$ quantitation in tumour interstitial fluid
LLC tumour cells ($1 \times 10^6$) were inoculated subcutaneously into EGFP mice. When the tumour volume reached approximately 200 mm³, 70 μg of DOX (intratumoural injection) was added. After 1 week, the

tumour interstitial fluid (TIF) was collected as previously described[41]. $Zn^{2+}$ concentration was determined using an AmpliteFluorimetric Zinc Ion Quantitation Kit (AAT-19000, AAT Bioquest, CA, USA).

## Calcein transfer

A549 cells ($5 \times 10^4$) were seeded into individual wells of 12-well plates and labelled with CM-Dil. PCCAFs or hCAFs were loaded with $0.5\,\mu M$ calcein-AM (354216; Corning, NY, USA) at 37 °C for 30 min. Fibroblasts ($1 \times 10^5$) were added to A549 cells and co-cultured for the indicated times in DMEM + 10% FBS. Heptanol (2 mM) was used to block the gap junctions. After treatment, calcein fluorescence in tumour cells was determined by FACS.

## Determination of S100A4 by ELISA

LLC-GFP-luc tumour-bearing mice were treated with PBS or DOX as described above. Two days after the third DOX treatment, tumour tissues were collected, weighed, and cut into small pieces (2–3 mm) in cold PBS (300 µL PBS was added to 100 mg of tissue) to release extracellular protein on ice. Tumour pieces were briefly squeezed with a disposable syringe rubber plunger, and the supernatants were collected after centrifugation ($500 \times g$, 2 min, 4 °C). S100A4 in the supernatant was determined using a murine S100A4 ELISA kit (JLC2761; Shanghai Jingkang Biocompany, China).

## Statistical analysis

All bars or symbols in the graphs represent the mean ± standard error of at least three independent experiments with similar results. The results were analysed using unpaired two-tailed $t$-tests or Mann–Whitney test for two-group comparisons, one-way ANOVA with Tukey's post hoc analysis for multigroup comparisons, and two-way ANOVA for curve comparisons. Statistical analyses were performed using GraphPad Prism v8 (GraphPad Software, La Jolla, CA, USA). Patient pathological parameter analysis was performed using SPSS Statistics 21 (IBM, NY, USA). Contingency data were analysed using $\chi^2$ tests. Survival curves were analysed using the log-rank test. Correlations were evaluated using Pearson's correlation analysis. $P < 0.05$ was considered statistically significant.

## Reporting summary

Further information on research design is available in the Nature Research Reporting Summary linked to this article.

## Data availability

The raw sequence data reported in this paper have been deposited in the Genome Sequence Archive at the National Genomics Data Center, China National Center for Bioinformation/Beijing Institute of Genomics, Chinese Academy of Sciences, under accession number CRA004556. Single-cell sequencing datasets of human lung adenocarcinoma were downloaded from NCBI's Gene Expression Omnibus (GSE123904), which has been previously published[38]. The data of supplementary Fig. 8g were derived from Kaplan–Meier plotter (kmplot.com/analysis) using publicly available datasets GSE14814, GSE29013 and datasets stored in caARRAY [https://wiki.nci.nih.gov/display/caArray2/caArray+Directory] for lung cancer, GSE62254 for gastric cancer, datasets stored in TCGA [https://gdc.cancer.gov] for ovarian cancer, and GSE1456, GSE16446, GSE16716, GSE20271, GSE22093, GSE3494, GSE37946, GSE45255, GSE69031 for breast cancer. Source data are provided in this paper. The remaining data are available within the Article, Supplementary Information or Source Data file. Source data are provided with this paper.

## Code availability

Code used for data processing, analysis, and figure generation in this study is available at: https://github.com/dekanglv/Zip1-Fibroblasts-Single-Cell-2022.

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

## Acknowledgements

This work was supported by grants from the National Key Research and Development Programme of China [2021YFA1201102 to Z.Q.], the National Natural Science Foundation of China [grant number 82073231 to Z.Q., 81902336 to C.N., 81602200 to D.L.], and the Key Project of Medical Science and Technology of Henan Province [grant number SB201902019 to C.N.]. We thank Hongzhang Deng, Fazhan Wang, Yan Chen, and Yongjuan Li for their helpful discussions. We thank Chunhui Wang for her critical comments. We would like to thank Dr. R. Phillips of Insight Editing London and Editage (www.editage.cn) for English language editing.

## Author contributions

Conceptualisation: Z.Q., C.N., and D.L. Methodology: C.N., D.L., X.L., X.Y., J.W., X.D., K.Z., Y.Y. and L. Zhang. Investigation: C.N., D.L., X.L., J.L., L.W., L. Zhu, C.S. and M.W. Visualisation: C.N., X.L., and J.W. Supervision: Z.Q. Writing—original draft: C.N. and D.L. Writing—review and editing: Z.Q., C.N., D.L., Z.L. and M.W.

## Competing interests

The authors declare no competing interests.
