## [Peer Review File · Nature Communications]

ZIP1+ fibroblasts protect lung cancer against chemotherapy via connexin-43 mediated intercellular Zn²⁺ transferEditorial Note: This manuscript has been previously reviewed at another journal that is not operating a transparent peer review scheme. This document only contains reviewer comments and rebuttal letters for versions considered at Nature Communications.

REVIEWER COMMENTS

Reviewer #1 (Remarks to the Author):

In the revised version the authors have responded to many of my comments and the manuscript is overall improved. However, the (very interesting) main claim is still not clearly supported by the data, and some of the data are still presented in a confusing manner.

Several specific concerns also remain and need to be addressed before this manuscript can be accepted:

1. In response to my comments the authors tested whether chemotherapy induces the expression of Zip1 in CAFs, and the results demonstrated that this was not the case. In fact, the new data show that Dox treatment down regulated the expression of Zip1 (new western blot in Fig. 7, not quantified). The authors now suggest that secreted S100A4 may be responsible for Zip1 upregulation (rather than a direct effect of chemotherapy). This may be the case. However, they state in the revised text that “After screening, we found that S100A4..” What screening was performed? Why was S100A4 a candidate? This needs to be clearly explained.

The authors now show in vitro that exogenous S100A4 (not explained in the text) can induce Zip1 expression in fibroblasts. But what is the cellular source of S100A4 in vivo? Based on new Figure 7j, cancer cells upregulate secretion of S100A4 following chemotherapy. To mechanistically tie up these new observations, the authors should show that CM from Dox-treated tumor cells can upregulate Zip1 in fibroblasts, and that targeting of S100A4 (e.g with siRNA) inhibits this upregulation.

2. Fig S1a: what are the arrows pointing at? Not clear and not explained in legends.

3. In response to my comment that western blots should be quantified, the authors responded that “For limited space, we did not include the semi-quantification results of western blots”. This is not an acceptable response. Quantifications can be included in supplementary information if space is limiting, but they should be performed. This is true also for the new western blots in Fig. 7.

4. Figure 2i: what is the arrow in the right panel pointing at? There is no explanation in the legends.

5. In response to my previous comment #14 regarding Fig. 4e, the authors responded: “We are sorry for the confusion. We have reviewed the origin Fig. 4e to clearly show a visible difference”. I am not sure what the authors mean by “reviewed the origin”. However, in the figure it still seems that TPEN does not inhibit the expression of cx43, whereas HEPT does. This too would be less confusing if WB were quantified.

6. Although the authors responded to my request for English editing by stating that they have done so, the manuscript is still poorly written and needs to be corrected for English grammar in some parts. For example (out of multiple others): We propose that ZIP1+S100A4+CX43high CAF may be called as zinc412 transport CAF (zCAF) that absorbs and transfers Zn²⁺ to neighboring cancer cells through gap junctions. Except for zCAF, our results indicate that ZIP1 might also (be) expressed in fibroblasts..”

Reviewer #2 (Remarks to the Author):

Ni et al. revised most of concerns raised by this reviewer, although some of important questions

remind unclear and they left them as speculative thoughts in discussion. Nevertheless, they indeed identified a unique population of CAF expressing ZIP1, that may directly control cancer cells to become chemoresistance by zinc influx through functional gap junctions which upregulates ABCB1 for drug extrusion. They have successfully revised and modify the previous version of manuscript which sounds improved, so that from this reviewer there is no further requests and comments for publication.

Reviewer #3 (Remarks to the Author):

The authors have responded appropriately to all of the reviewers' comments including providing additional data. The manuscript and study would have been significantly improved if they had other preclinical models besides the LLC model- and there are several such cell line syngeneic models available that are derived from genetic engineered mouse models. They have increased their data on human patient tumor responses and that is particularly important.

Reviewer #4 (Remarks to the Author): new reviewer with expertise in CAFs, scRNAseq

In this manuscript, Ni et al., describe a unique Zip1+ CAF population with the ability to communicate with tumor cells via connexin-43, enabling Zn²⁺ transfer and facilitating chemoresistance. They conducted single cell analyses of stromal populations derived from subcutaneous LLC tumors treated with Dox and demonstrate the enrichment of Zip1+ fibroblasts in response to treatment. They supplement their in-silico findings with experimental substantiation and show the relevance of Zip1 expressing fibroblasts in determining clinical outcomes. This is an important and relatively under-investigated area of research where the authors have done a commendable assessment of interactions between CAFs and cancer cells. My main concerns relate to the analysis of single cell data, annotation of CAF subtypes and relevance to human single cell RNA seq data in lung cancer. Please see major and minor comments detailed below:

Major comments:

1. Based on the clustering resolution it seems that C1 CAFs in Figure 1c most likely represent an activation program rather than a distinct CAF cluster (clustering resolutions are subjective and in this case the boundaries seem to be arbitrary based on the choice of the resolution parameter). Can you comment on which genes are expressed in this cluster (disregarding their specificity)? Are they shared with Zip1+ CAFs? In such cases it is useful to use topic modeling approaches to show changes in activation programs (refer cNMF Kotliar et al., eLife 2019)
2. Line 126: The evidence in support of the ubiquity of Zip1+ fibroblasts across lung and pancreatic cancer are weak. Can you illustrate this by using publicly available mouse scRNA seq datasets, for example, Dominguez & Muller et al., Cancer Discovery 2020 (pdac), Bartoschek et al., Nature Communications 2018 (breast cancer), Zilionis et al., Immunity 2019 (lung cancer)? It would be important to map the Zip1+ fibroblast signature derived from your dataset to these single cell datasets to determine whether these CAF subsets truly exist in multiple indications.
3. It is unclear how these CAF subsets relate to previously described Il1 CAFs, TGF-beta driven CAFs etc. It is important to show which signaling pathways are activated in these subsets. Please use Progeny to determine signaling pathway responsive genes. It will be interesting to see if the Zip1+ fibroblasts are driven by a completely different pathway.
4. RNA velocity results suggest that these Zip1+ CAFs are able to give rise to cluster 2 CAFs – how does this compare to previous observations where Dpt+ universal fibroblasts give rise to activated fibroblasts such as Lrrc15+ CAFs in different cancer indications (Buechler & Pradhan 2021). Do Zip1+

fibroblasts express other markers of stemness that would pinpoint to this particular function? It is important to mention here which genes are upregulated in the transition from Zip1+/ Cluster 0 to cluster 2 and Cluster A0 to A01.

5. In Maynard et al., Cell 2020, the authors study therapy induced adaptation in advanced NSCLC using single cell RNA seq. They show that cancer cells surviving on therapy progressive disease upregulate gap-junction pathways. As such it would be important to check whether Zip1+ fibroblast signature from this manuscript's dataset maps to CAF subtypes in the human NSCLC data.

6. For Cluster 4 expressing both macrophage and fibroblast markers, can you check if this is truly the case or these represent doublets? If so, please use scrublet to remove these cells.

Minor comments:

1. Figure 1d, S1g: Add scale/ legend. Alternatively, plot DotPlot depicting both gene expression levels and percentage of cells expressing the gene in a cluster.

2. Figure 1f, S1e: Please change to a faceted bar graph for PBS versus Dox treated samples for ease in interpreting the visualization. Ensure that colors for clusters in the bar graph match the UMAP colors in Figure 1c.

3. Figure 1h: Please denote the RNA velocity direction clearly in the figure panel as it is challenging to see the direction of the arrow. Adding cluster labels will help with clarity in visualization.

Point-by-point response

REVIEWER COMMENTS

Reviewer #1 (Remarks to the Author):

In the revised version the authors have responded to many of my comments and the manuscript is overall improved. However, the (very interesting) main claim is still not clearly supported by the data, and some of the data are still presented in a confusing manner.

Several specific concerns also remain and need to be addressed before this manuscript can be accepted:

1. In response to my comments the authors tested whether chemotherapy induces the expression of Zip1 in CAFs, and the results demonstrated that this was not the case. In fact, the new data show that Dox treatment down regulated the expression of Zip1 (new western blot in Fig. 7, not quantified). The authors now suggest that secreted S100A4 may be responsible for Zip1 upregulation (rather than a direct effect of chemotherapy). This may be the case. However, they state in the revised text that “After screening, we found that S100A4..” What screening was performed? Why was S100A4 a candidate? This needs to be clearly explained.

The authors now show in vitro that exogenous S100A4 (not explained in the text) can induce Zip1 expression in fibroblasts. But what is the cellular source of S100A4 in vivo? Based on new Figure 7j, cancer cells upregulate secretion of S100A4 following chemotherapy. To mechanistically tie up these new observations, the authors should show that CM from Dox-treated tumor cells can upregulate Zip1 in fibroblasts, and that targeting of S100A4 (e.g with siRNA) inhibits this upregulation.

Response: Thanks for the reviewer’s insightful comments and suggestions. Indeed, we have examined the effects of several secretory factors including TGFβ, TNFα, IFNγ and S100A4 on ZIP1 expression in fibroblasts. Only S100A4 could upregulate ZIP1 expression in fibroblasts. For clarity and focus reason, we did not display the results for TGFβ, TNFα and IFNγ and deleted “After screening”. We have added “Other’s and our own previous studies demonstrated that S100A4 often increases in tissues under stress^{1, 2, 3, 4}” in the revised manuscript to explain why we chose S100A4 as a candidate (**Page 15 line 313-314**).

As the reviewer suggested, we performed experiments to examine whether DOX treatment increases the release of S100A4 by tumor cells, and whether S100A4 in culture medium from DOX-treated tumor cells can upregulate ZIP1 in fibroblasts. The results showed that DOX treatment increased the release of S100A4 into the supernatants of tumor cells (**Fig. 7c, Fig. S7c**). Compared to the control medium, ZIP1 expression in fibroblasts was upregulated by the culture medium from DOX-treated tumor cells, which could be reversed by anti-S100A4 antibody 3B11 (**Fig. 7d, Fig. S7d**) (**Page 15 line 314-317**).

As the reviewer suggested, we have quantified western-blotting results in **Fig. 7**.

2. Fig S1a: what are the arrows pointing at? Not clear and not explained in legends.

Response: We apologize for the confusion. The arrows mean DOX injection at day 0, 2 and 4. We have explained the arrows in the legend of **Fig S1a (Supplementary figures and figure legends)**.

3. *In response to my comment that western blots should be quantified, the authors responded that “For limited space, we did not include the semi-quantification results of western blots”. This is not an acceptable response. Quantifications can be included in supplementary information if space is limiting, but they should be performed. This is true also for the new western blots in Fig. 7.*

Response: Thanks for the reviewer’s kind suggestions. As the reviewer suggested, we have quantified the western-blotting results including **Fig. 7** in our revised manuscript, and the results are included in supplementary information.

4. *Figure 2i: what is the arrow in the right panel pointing at? There is no explanation in the legends.*

Response: We apologize for the confusion. The arrow points at the position of hCAF in the graph. We have explained the arrow in the legend of **Figure 2i**.

5. *In response to my previous comment #14 regarding Fig. 4e, the authors responded: “We are sorry for the confusion. We have reviewed the origin Fig. 4e to clearly show a visible difference”. I am not sure what the authors mean by “reviewed the origin”. However, in the figure it still seems that TPEN does not inhibit the expression of cx43, whereas HEPT does. This too would be less confusing if WB were quantified.*

Response: Thanks for the reviewer’s kind comments and suggestions. For **Fig. 4e**, we adjusted the brightness and contrast of the raw picture to reveal group difference. As the reviewer suggested, we have quantified western blot in **Fig. 4e (Fig. S4f)**.

6. *Although the authors responded to my request for English editing by stating that they have done so, the manuscript is still poorly written and needs to be corrected for English grammar in some parts. For example (out of multiple others): We propose that ZIP1+S100A4+CX43high CAF may be called as zinc412 transport CAF (zCAF) that absorbs and transfers Zn²⁺ to neighboring cancer cells through gap junctions. Except for zCAF, our results indicate that ZIP1 might also (be) expressed in fibroblasts.”*

Response: Thanks for the reviewer’s kind suggestions. We have asked a professional language editing company (Editage, www.editage.cn) to improve the English writing of our manuscript again. Hopefully, current version of the manuscript is error free and suitable for publication.

Reviewer #2 (Remarks to the Author):

Ni et al. revised most of concerns raised by this reviewer, although some of important questions remind unclear and they left them as speculative thoughts in discussion. Nevertheless, they indeed identified a unique population of CAF expressing ZIP1, that may directly control cancer cells to become chemoresistance by zinc influx through functional gap junctions which upregulates ABCB1 for drug extrusion. They have successfully revised and modify the previous version of manuscript which sounds improved, so that from this reviewer there is no further requests and comments for publication.

Response: We deeply appreciate the reviewer’s kind comments.

Reviewer #3 (Remarks to the Author):

The authors have responded appropriately to all of the reviewers' comments including providing additional data. The manuscript and study would have been significantly improved if they had other preclinical models besides the LLC model- and there are several such cell line syngeneic models available that are derived from genetic engineered mouse models. They have increased their data on human patient tumor responses and that is particularly important.

Response: Thanks for the reviewer's kind comments. We will consider examining more preclinical lung cancer models in our future studies.

Reviewer #4 (Remarks to the Author): new reviewer with expertise in CAFs, scRNAseq

In this manuscript, Ni et al., describe a unique Zip1+ CAF population with the ability to communicate with tumor cells via connexin-43, enabling Zn2+ transfer and facilitating chemoresistance. They conducted single cell analyses of stromal populations derived from subcutaneous LLC tumors treated with Dox and demonstrate the enrichment of Zip1+ fibroblasts in response to treatment. They supplement their in-silico findings with experimental substantiation and show the relevance of Zip1 expressing fibroblasts in determining clinical outcomes. This is an important and relatively under-investigated area of research where the authors have done a commendable assessment of interactions between CAFs and cancer cells. My main concerns relate to the analysis of single cell data, annotation of CAF subtypes and relevance to human single cell RNA seq data in lung cancer. Please see major and minor comments detailed below:

Major comments:

1. Based on the clustering resolution it seems that C1 CAFs in Figure 1c most likely represent an activation program rather than a distinct CAF cluster (clustering resolutions are subjective and in this case the boundaries seem to be arbitrary based on the choice of the resolution parameter). Can you comment on which genes are expressed in this cluster (disregarding their specificity)? Are they shared with Zip1+ CAFs? In such cases it is useful to use topic modeling approaches to show changes in activation programs (refer cNMF Kotliar et al., eLife 2019)

Response: Thanks for the reviewer's insightful comments. As the reviewer suggested, we performed cNMF analysis of C0-C3. We identified the top 100 genes for each GEP (**Table S3**) and performed a GO (Gene Ontology) enrichment analysis to explore the function of each GEP (**Table S4**). Relating to the clusters and programs, we found that of the four clusters, cluster 0 was strongly enriched for developmental GEP1, cluster 1 was strongly enriched for metabolic GEP2, cluster 2 was strongly enriched for matrix GEP3, and cluster 3 was strongly enriched for proliferating GEP4, further supporting that they are distinct clusters (**Fig. S1i**). Although there were no specific markers, cluster 1 highly expressed metabolism-related genes, such as *Gchfr*, *Mif*, and *Cox8a*, that were shared with other clusters (**Fig. S1j**). (**Results, page 6-7 line 125-133. Methods, page 29 line 626-630**).

2. Line 126: The evidence in support of the ubiquity of Zip1+ fibroblasts across lung and pancreatic cancer are weak. Can you illustrate this by using publicly available mouse scRNA seq datasets, for example, Dominguez & Muller et al., Cancer Discovery 2020 (pdac), Bartoschek et al., Nature

Communications 2018 (breast cancer), Zilionis et al., Immunity 2019 (lung cancer)? It would be important to map the Zip1+ fibroblast signature derived from your dataset to these single cell datasets to determine whether these CAF subsets truly exist in multiple indications.

Response: Thanks for the reviewer's kind comments. We agree that it is important to mount bioinformatic and experimental evidences in support of the ubiquity of ZIP1+ fibroblasts across cancer types. However, whether ZIP1+ fibroblasts play similar functions in different cancer types is unclear, and needs careful study individually. Therefore, in this study, we majorly focus our study of ZIP1+ fibroblasts in lung cancer. We have mapped the ZIP1+ fibroblast signature to CAF subtypes in the human NSCLC data (**Fig. S8, Table S6-7, page 15-16 line 330-346**). We found that there is a corresponding ZIP1+ fibroblast subset in the human NSCLC (**Fig. S8**). The immunostaining of ZIP1+ fibroblasts in transplanted Pan02 tumor further supports their existence in mouse tumor model, and is a good implication for further study of ZIP1+ fibroblasts in pancreatic cancer as well as other cancer types. As the reviewer suggested, we will consider the research of ZIP1+ fibroblast function in other tumors in the future.

3. It is unclear how these CAF subsets relate to previously described I1 CAFs, TGF-beta driven CAFs etc. It is important to show which signaling pathways are activated in these subsets. Please use Progeny to determine signaling pathway responsive genes. It will be interesting to see if the Zip1+ fibroblasts are driven by a completely different pathway.

Response: Thanks for the reviewer's insightful comments and suggestions. As the reviewer suggested, we have performed PROGENY to determine signaling pathway responsive genes in each cluster (**Methods, page 29 line 632-635**). The PI3K signalling pathway was active in cluster 0 *Zip1*⁺ CAFs, while the TGFβ signalling pathway was active in cluster 6 and 7 myofibroblasts, the JAK-STAT signalling pathway was active in cluster 4 fibrocytes, and the VEGF signalling pathway was active in cluster 2 (**Fig. S1k**). It has been reported that TGFβ stimulates myofibroblast phenotype and IL-1 (activating JAK-STAT) promotes inflammatory fibroblasts^{5, 6}. Therefore, cluster 0 *Zip1*⁺ CAFs might be driven by a distinct PI3K signalling pathway compared to other clusters. (**Page 7 line 133-140**).

4. RNA velocity results suggest that these Zip1+ CAFs are able to give rise to cluster 2 CAFs – how does this compare to previous observations where Dpt+ universal fibroblasts give rise to activated fibroblasts such as Lrrc15+ CAFs in different cancer indications (Buechler & Pradhan 2021). Do Zip1+ fibroblasts express other markers of stemness that would pinpoint to this particular function? It is important to mention here which genes are upregulated in the transition from Zip1+ Cluster 0 to cluster 2 and Cluster A0 to A01.

Response: Thanks for the reviewer's insightful comments. Cluster 0 *Zip1*⁺ CAFs moderately expressed *Dpt* and cluster 2 CAFs expressed higher *Dpt* (**Fig. S1h**), indicating these fibroblasts are similar to *Dpt*⁺ universal fibroblasts. Cluster 0 *Zip1*⁺ CAFs expressed *Notch2*, which has been reported to be critical for the maintenance of cell stemness in hematopoietic cells and neural stem cells^{7, 8}. (**Page 6 line 123-125**). As the reviewer suggested, we evaluated gene upregulation during cluster transition. For example, *Plod2*, *Nnmt* and *Col3a1* were upregulated in transition from Cluster 0 to Cluster 2, while *Spry2*, *Mt2* and *Mt1* were upregulated in transition within cluster 0 (**Fig. S1l-o, Table S5**). (**Results, page 7 line 145-147, Methods, page 29 line 621-624**).

5. In Maynard et al., Cell 2020, the authors study therapy induced adaptation in advanced NSCLC using single cell RNA seq. They show that cancer cells surviving on therapy progressive disease upregulate gap-junction pathways. As such it would be important to check whether Zip1+ fibroblast signature from this manuscript's dataset maps to CAF subtypes in the human NSCLC data.

Response: Thanks for the reviewer's insightful comments and suggestions. We have discussed the study of Maynard et al. in **Discussion**. We added "In support of our results, Maynard et al. reported that cancer cells surviving on therapy progressive disease upregulated the gap junction pathway in lung cancer patients with targeted therapy"⁹ in the revised manuscript. (**Page 20 line 423-425**).

We agree to the reviewer that it is important to check whether Zip1+ fibroblast signature maps to CAF subtypes in the human NSCLC data. Indeed, we have performed these analyses in our manuscript (**Fig. S8, Table S6-7, page 15-16 line 330-346**). We found that there is a corresponding ZIP1+ fibroblast subset in the human NSCLC.

6. For Cluster 4 expressing both macrophage and fibroblast markers, can you check if this is truly the case or these represent doublets? If so, please use scrublet to remove these cells.

Response: Thanks for the reviewer's comments and suggestions. We performed Scrublet to check doublets in our dataset. We identified overall 5.2% doublets in total cells and especially 7.6% in Cluster 4 (**Figure R1a**). We compared the expression of feature genes of Cluster 4 (*Itgm, CD14, Lyz2, Dcn, Fn1, Sparc, Spp1*) in original dataset and dataset removed doublets. The results showed that the expression pattern of feature genes (*Itgm, CD14, Lyz2, Dcn, Fn1, Sparc, Spp1*) in the two datasets was almost the same (**Figure R1b**), suggesting that the expression of both macrophage and fibroblast markers by cluster 4 was not caused by doublets. A recent study also identified fibroblasts expressed both fibroblast and macrophage markers and proposed macrophage-myofibroblast transition mechanism for generation of these cells¹⁰ (**Results, page 5 line 98-100**).

a

Cluster	Number of cells	Doublets	Doublet ratio
0	7681	718	9.30%
1	4154	57	1.30%
2	3779	73	1.90%
3	3082	118	3.80%
4	394	30	7.60%
5	390	25	6.40%
6	73	0	0%
7	64	1	1.50%
ALL	19617	1022	5.20%

b

Figure R1. Identification of doublets in Cluster 4 using scrublet. (a) Doublet number and ratio in each cluster. (b). Expression of feature genes of Cluster 4 in each cluster. For the detection of potential doublet cells, we applied the scrublet v.0.2.3 pipeline to each subset with parameters (min_count = 3, min_cells = 3, vscore_percentile= 85, n_prin_comps=30, expected_doublet_rate = 0.06, sim_doublet_ratio = 2, n_neighbours = 50, log_transform=True) for doublet score calculation. Cells with doublet score over 0.25 are annotated as detected doublets. We detected 5.2% potential doublet cells in the whole dataset.

Minor comments:

1. *Figure 1d, S1g: Add scale/ legend. Alternatively, plot DotPlot depicting both gene expression levels and percentage of cells expressing the gene in a cluster.*

Response: Thanks for the reviewer's kind suggestions. As the reviewer suggested, we have added scale/legend for **Figure 1d, Figure S1g**.

2. *Figure 1f, S1e: Please change to a faceted bar graph for PBS versus Dox treated samples for ease in interpreting the visualization. Ensure that colors for clusters in the bar graph match the UMAP colors in Figure 1c.*

Response: Thanks for the reviewer's kind suggestions. As the reviewer suggested, we have changed **Figure 1f, S1e** to a faceted bar graph.

3. *Figure 1h: Please denote the RNA velocity direction clearly in the figure panel as it is challenging to see the direction of the arrow. Adding cluster labels will help with clarity in visualization.*

Response: Thanks for the reviewer's kind suggestions. As the reviewer suggested, we have added cluster labels and denoted the RNA velocity direction in the figure panel (**Figure 1h**).

References

1. Chen L, *et al.* S100A4 promotes liver fibrosis via activation of hepatic stellate cells. *Journal of hepatology* **62**, 156-164 (2015).
2. Li Y, *et al.* S100A4(+) Macrophages Are Necessary for Pulmonary Fibrosis by Activating Lung Fibroblasts. *Frontiers in immunology* **9**, 1776 (2018).
3. Liu Y, *et al.* Extracellular ATP drives breast cancer cell migration and metastasis via S100A4 production by cancer cells and fibroblasts. *Cancer letters* **430**, 1-10 (2018).
4. Li Z, Li Y, Liu S, Qin Z. Extracellular S100A4 as a key player in fibrotic diseases. *J Cell Mol Med* **24**, 5973-5983 (2020).
5. Biffi G, Tuveson D. Diversity and Biology of Cancer-Associated Fibroblasts. *Physiological reviews* **101**, 147-176 (2021).

6. Biffi G, *et al.* IL1-Induced JAK/STAT Signaling Is Antagonized by TGF β to Shape CAF Heterogeneity in Pancreatic Ductal Adenocarcinoma. *Cancer discovery* **9**, 282–301 (2019).
7. Varnum-Finney B, Halasz LM, Sun M, Gridley T, Radtke F, Bernstein ID. Notch2 governs the rate of generation of mouse long- and short-term repopulating stem cells. *The Journal of clinical investigation* **121**, 1207–1216 (2011).
8. Zhang R, *et al.* Id4 Downstream of Notch2 Maintains Neural Stem Cell Quiescence in the Adult Hippocampus. *Cell reports* **28**, 1485–1498.e1486 (2019).
9. Maynard A, *et al.* Therapy-Induced Evolution of Human Lung Cancer Revealed by Single-Cell RNA Sequencing. *Cell* **182**, 1232–1251.e1222 (2020).
10. Tang PC, *et al.* Smad3 Promotes Cancer-Associated Fibroblasts Generation via Macrophage-Myofibroblast Transition. *Advanced science (Weinheim, Baden-Wuerttemberg, Germany)* **9**, e2101235 (2022).

REVIEWERS' COMMENTS

Reviewer #1 (Remarks to the Author):

In this revised version, the authors have addressed my comments and the manuscript is much improved.

One last comment remain before the paper can be accepted:

In new Fig. 7C, there is no loading control (all other blots have b-actin). This should be included. In fact, how was the gel quantified without loading control? In principle, quantification should be relative to controls, but also to loading control. How quantifications were performed is not explained in legends or figures and it should be clearly stated.

Reviewer #4 (Remarks to the Author):

Ni et al., have adequately addressed my concerns regarding the bioinformatics analysis and interpretation of the data. While it is not entirely clear whether this CAF can be annotated as a new CAF type, experimental findings do suggest that Zip1+ CAFs emerge uniquely in response to chemotherapy. In light of improved analyses and successful modification of the manuscript, there are no further comments for publication.

Point-by-point response

REVIEWERS' COMMENTS

Reviewer #1 (Remarks to the Author):

In this revised version, the authors have addressed my comments and the manuscript is much improved.

One last comment remain before the paper can be accepted:

In new Fig. 7C, there is no loading control (all other blots have b-actin). This should be included. In fact, how was the gel quantified without loading control? In principle, quantification should be relative to controls, but also to loading control. How quantifications were performed is not explained in legends or figures and it should be clearly stated.

Response: Thanks for the reviewer's kind comments and suggestions. β -actin is not an available loading control for detecting secretory proteins in culture medium (CM) by western-blotting. In **Fig. 7c**, S100A4 levels were evaluated in the same volume of CM from different treatment groups. The same number of tumour cells were seeded and pre-treated with DOX (1 μ M) for 6 h. Twenty-four hours after changing fresh medium, CMs were collected. The same volume of CM from different treatment groups was analysed by Western-blotting. The experiment was repeated independently with similar results. Quantification was performed relative to controls. This has been explained in the legends of **Fig. 7c (Page 52 line 1229-1233)** and **Fig. S7c**. All other blot quantifications have used β -actin as internal loading control.

Reviewer #4 (Remarks to the Author):

Ni et al., have adequately addressed my concerns regarding the bioinformatics analysis and interpretation of the data. While it is not entirely clear whether this CAF can be annotated as a new CAF type, experimental findings do suggest that Zip1+ CAFs emerge uniquely in response to chemotherapy. In light of improved analyses and successful modification of the manuscript, there are no further comments for publication.

Response: We deeply appreciate the reviewer's kind comments.